# A *Salmonella* Typhi RNA thermosensor regulates virulence factors and innate immune evasion in response to host temperature

**Susan M. Brewer**[1], **Christian Twittenhoff**[2], **Jens Kortmann**[3], **Sky W. Brubaker**[1], **Jared Honeycutt**[1], **Liliana Moura Massis**[1], **Trung H. M. Pham**[1], **Franz Narberhaus**[2], **Denise M. Monack**[1‡]*

**1** Department of Microbiology and Immunology, Stanford University School of Medicine, Stanford, California, United States of America, **2** Microbial Biology, Ruhr University, Bochum, Germany, **3** Genentech, Inc., South San Francisco, California, United States of America

‡ Lead contact
* dmonack@stanford.edu

**Data Availability Statement:** All relevant data are within the manuscript and its Supporting Information files.

## Abstract

Sensing and responding to environmental signals is critical for bacterial pathogens to successfully infect and persist within hosts. Many bacterial pathogens sense temperature as an indication they have entered a new host and must alter their virulence factor expression to evade immune detection. Using secondary structure prediction, we identified an RNA thermosensor (RNAT) in the 5' untranslated region (UTR) of *tviA* encoded by the typhoid fever-causing bacterium *Salmonella enterica* serovar Typhi (*S.* Typhi). Importantly, *tviA* is a transcriptional regulator of the critical virulence factors Vi capsule, flagellin, and type III secretion system-1 expression. By introducing point mutations to alter the mRNA secondary structure, we demonstrate that the 5' UTR of *tviA* contains a functional RNAT using *in vitro* expression, structure probing, and ribosome binding methods. Mutational inhibition of the RNAT in *S.* Typhi causes aberrant virulence factor expression, leading to enhanced innate immune responses during infection. In conclusion, we show that *S.* Typhi regulates virulence factor expression through an RNAT in the 5' UTR of *tviA*. Our findings demonstrate that limiting inflammation through RNAT-dependent regulation in response to host body temperature is important for *S.* Typhi's "stealthy" pathogenesis.

## Author summary

*Salmonella enterica* serovar Typhi (*S.* Typhi) is a human-restricted bacterial pathogen that causes 11–21 million new cases of typhoid fever worldwide each year. During acute infection, *S.* Typhi evades immune detection and rapidly spreads systemically, which has earned it the nickname of a "stealth" pathogen. The transcriptional regulator TviA is unique to *S.* Typhi and critical for this stealthy phenotype because it represses the expression of major pathogen-associated molecular patterns and induces the important

**Funding:** Research reported in this publication was supported by grant R01-AI116059 from the National Institute of Allergy and Infectious Diseases, United States (to D.M.M.), Paul Allen Stanford Discovery Center on Systems Modeling of Infection (to D.M.M. https://alleninstitute.org/what-we-do/frontiers-group/discovery-centers/allen-discovery-center-stanford-university/), DFG project NA 240/10-2 from the German Research Foundation (to F.N.), and the Stanford Graduate Fellowship (to S.M.B. https://vpge.stanford.edu/fellowships-funding/sgf). The funders had no role in study design, data collection and analysis, decision to publish, or preparation of the manuscript.

**Competing interests:** The authors have declared that no competing interests exist.

virulence factor Vi capsule, which enable *S.* Typhi to evade immune detection. We show that *S.* Typhi regulates virulence factor expression in response to host body temperature via an RNA thermosensor (RNAT) located in the 5' untranslated region of *tviA*, which controls production of TviA protein in response to temperature fluctuations. Mutational perturbation of the *tviA* RNAT causes aberrant virulence factor expression in response to temperature and reveals that a functional RNAT is required for *S.* Typhi to evade innate immune detection and activation. Our work is the first to describe an RNAT in a critical virulence factor transcriptional regulator in *S.* Typhi and demonstrates the important role that temperature-sensing plays in *S.* Typhi pathogenesis.

## Introduction

Microbes use environmental cues to modulate their expression profiles to adapt to new niches. Many bacterial pathogens sense temperature as an indication they have entered a new host [1]. At the human body temperature of 37˚C, many pathogens upregulate key virulence factors to defend against the immune response and modify or downregulate expression of molecules that may alert the host to their presence. Some bacterial pathogens have evolved RNA thermosensors (RNATs) to rapidly modulate their virulence factor expression in response to temperature [2]. RNATs are typically found in the 5' untranslated region (UTR) of bacterial genes and post-transcriptionally regulate their translation [2]. For example, at lower temperatures, the mRNA secondary structure obscures the Shine-Dalgarno (SD) region and/or the translation start codon (AUG), preventing ribosomes from binding and translating the downstream gene product (S1A Fig). As the temperature increases, the hydrogen bonds between the nucleotides melt, opening up the structure and enabling ribosome access and translation (S1A Fig) [2]. Using post-transcriptional RNAT-mediated thermoregulation enables bacterial pathogens to rapidly respond to a temperature increase by immediately translating critical transcription factors that impact pathogenesis. Although previous studies have identified RNATs in the 5' UTR of transcription regulators that control the expression of critical virulence factors, the role of RNATs in *Salmonella* pathogenesis has been understudied [2]. Indeed, the impact of temperature on *Salmonella enterica* serovar Typhi (*S.* Typhi), the causative agent of typhoid fever, has not been extensively explored.

*S.* Typhi is a human-restricted bacterial pathogen that causes an estimated 11–21 million new cases of typhoid fever worldwide annually [3]. Symptoms of this enteric fever manifest as headache, abdominal pain, and high fever. Although infection occurs through the fecal-oral route by ingesting contaminated food or water, *S.* Typhi transiently occupies the gastrointestinal niche, causing minimal inflammation, before rapidly disseminating to systemic sites where it can persist chronically [4,5]. In stark contrast to the highly related pathogen *S.* Typhimurium, which induces intestinal inflammation, leading to a self-limited gastroenteritis, *S.* Typhi evades immune detection during acute infection and spreads systemically, earning it the nickname of a "stealth" pathogen [4,5]. Despite sharing approximately 89% of genes, and many common virulence factors, the pathologies of the diseases caused by *S.* Typhi and *S.* Typhimurium are strikingly different [6]. Studying the genetic differences between these highly related strains has provided insight into how these bacterial pathogens cause such disparate diseases.

*S.* Typhi encodes a virulence factor locus that is absent in *S.* Typhimurium called *viaB*, which consists of the genes to synthesize (*tviBCDE*) and export (*vexABCDE*) the virulence factor Vi capsule [7,8]. Vi capsule is an extracellular polysaccharide coating that protects *S.* Typhi from innate immune responses by preventing TLR4 activation, complement deposition, and

neutrophil recruitment [9–12]. *S.* Typhi strains that lack Vi capsule are significantly attenuated, indicating the importance of this virulence factor in typhoid pathogenesis [13].

Expression of Vi capsule requires the transcriptional regulator *tviA*, another *viaB* locus gene, whose transcription is controlled by environmental osmolarity [14,15]. Under high salt conditions, such as those in the intestinal lumen, *tviA* transcription is suppressed, and Vi capsule is not expressed (S1B Fig) [15–17]. Importantly, TviA also negatively regulates the expression of flagellin, type three secretion system (T3SS)-1, and chemotaxis genes, which are important for motility and invasion of host cells [15]. Thus, in the absence of TviA, these molecules are fully expressed, and *S.* Typhi has a motile, invasive phenotype. As *S.* Typhi invades intestinal epithelial cells, the osmolarity decreases and transcription of *tviA* is induced, leading to Vi capsule expression and repression of flagellin, T3SS-1, and chemotaxis genes (S1B Fig) [15,18]. Repression of flagellin and T3SS-1 is critical upon tissue invasion because these are major pathogen-associated molecular patterns (PAMPs) that activate innate immunity. TLR5 recognizes flagellin, resulting in proinflammatory cytokine and chemokine secretion [19,20]. Cytosolic flagellin and T3SS-1 are recognized by NAIP, which activates the NLRC4 inflammasome, releasing the pro-inflammatory cytokines IL-1β and IL-18 and destroying *S.* Typhi's replicative niche via pyroptotic cell death [21–29]. Thus, in combination with Vi capsule expression, TviA-mediated suppression of flagellin and T3SS-1 is critical for *S.* Typhi to evade recognition by and activation of the innate immune system following tissue invasion. Since current vaccines target Vi antigen, it is crucial that we understand how this major virulence factor is regulated in *S.* Typhi [4]. In this study, we explore the mechanisms by which *S.* Typhi modulates the expression of major virulence factors in response to temperature. We demonstrate the presence and function of an RNAT in the 5' UTR of *tviA*, which modulates virulence factor expression in *S.* Typhi and is required for evasion of host innate immune responses.

## Results

### Temperature alters expression of *S.* Typhi virulence factors

Since *S.* Typhi can be isolated from environmental reservoirs such as water, we hypothesized that *S.* Typhi may regulate important virulence factors, such as Vi capsule, in a temperature-dependent manner [30]. To test this, we cultured the common *S.* Typhi lab strain Ty2 at an "environmental" temperature of 23°C, followed by a 2 hour shift to the human host body temperature of 37°C and looked at Vi capsule expression via immunofluorescence. *S.* Typhi Vi capsule expression was minimal at 23°C (Fig 1A). In contrast, Vi capsule expression increased following a shift to 37°C (Fig 1A), suggesting temperature-dependent regulation of virulence factor expression. To quantify Vi capsule expression and determine how quickly this expression change occurs, we performed a shorter temperature shift experiment and measured Vi expression via flow cytometry (S2 Fig). A 30 minute exposure to 37°C led to a significant increase in Vi capsule expression in WT Ty2, which was dependent on *tviA* (Fig 1B and 1C). Since TviA regulates the expression of Vi capsule, we proposed that the mRNA expression of TviA may be regulated by temperature, explaining the increase in Vi capsule expression we observed. To test this idea, we measured *tviA* mRNA levels in WT and Δ*tviA S.* Typhi strains following a 30 minute shift to 37°C. Interestingly, *tviA* mRNA levels did not increase at 37°C (Fig 1D). These data demonstrate that *S.* Typhi rapidly changes Vi capsule expression at higher temperature. However, this change in Vi capsule expression is not regulated at the *tviA* transcript level.

### The *tviA* 5' UTR modulates translation in response to temperature

Several human pathogens utilize RNA thermosensors (RNATs) to post-transcriptionally regulate the translation of virulence transcription factors in response to host body temperature

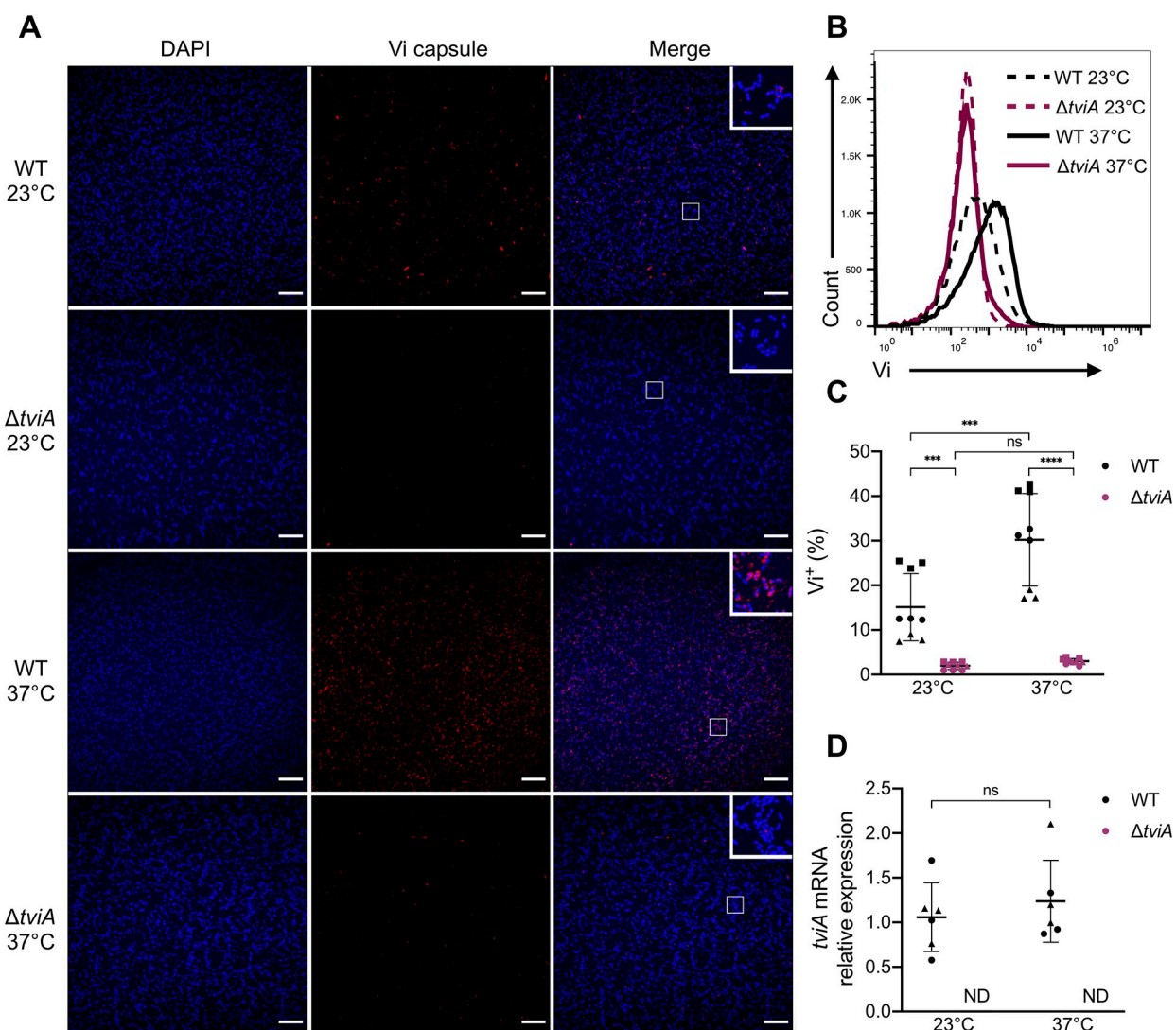

**Fig 1. *Salmonella* Typhi Vi capsule expression is elevated after shift from ambient to human body temperature.** A) Immunofluorescence microscopy images of wild-type (WT) or Vi capsule deficient (Δ*tviA*) *S.* Typhi Ty2 grown at 23°C (upper panels) or grown at 23°C and shifted to 37°C for 2 hours (lower panels). Bacteria were stained using Vi capsule-specific antibody (red) and DAPI counterstain (blue). Merged images show zoomed in insets of bacteria. Scale bar = 20 μm. B) Vi capsule expression measured by flow cytometry of wild-type (WT) or Δ*tviA* *S.* Typhi grown at 23°C or after a 30 minute shift to 37°C. C) Percentage of Vi⁺ WT or Δ*tviA* *S.* Typhi bacteria grown at 23°C or after a 30 minute shift to 37°C as measured by flow cytometry. D) Relative expression levels of *tviA* mRNA of WT and Δ*tviA* *S.* Typhi at ambient temperature (23°C) or following a shift to human body temperature (37°C) for 30 minutes. *tviA* mRNA levels normalized to 16S mRNA levels. ΔΔCt method used to determine relative expression compared to WT at 23°C. Data shown are representative of 3 (A and B) independent experiments or are the combined results of 3 (C) or 2 (D) independent experiments with triplicate samples for each condition. Symbol shape corresponds with datapoints from different independent experiments. Data are represented as mean ± SD. ND, not detected. NS, not significant. Significance calculated using two-way ANOVA with Tukey's correction. *** $p < 0.001$, **** $p < 0.0001$.

[31–35]. Since TviA is essential for Vi capsule expression and we did not observe changes in *tviA* mRNA levels following a temperature shift to 37°C (Fig 1D), we hypothesized that an RNAT may be present in the 5' UTR of *tviA*.

To investigate whether an RNAT may control *S.* Typhi's temperature-dependent virulence factor expression, we used Mfold to predict the secondary structure of the 5' UTR of *tviA* mRNA [36]. Analysis revealed that this leader region comprises a potential RNAT, in which the SD sequence (position 123 to 127 within the 5' UTR) partially pairs with four consecutive

uridines (fourU; position 89 to 92 within the 5' UTR) (Fig 2A). The 5' UTR is predicted to form a two stem-loop structure, with the 3'-most stem-loop harboring the promising fourU element. Moreover, this needle structure possesses destabilizing structural elements, such as an internal loop, a bulged adenine residue, and non-canonical base-pairings between the fourU element and the SD sequence. The predicted 5'-stem loop comprises a multi loop structure with two individual hairpins. These structural properties are consistent with those of previously described *cis*-acting RNATs, which have been shown to possess two hairpins with the 3'-most hairpin harboring the thermosensor element, bulged nucleotides or internal loops, and/or the fourU element participating in noncanonical base-pairing [31,32,35,37–41]. Thus, these predicted structural properties as well as the observed thermoregulation of Vi capsule expression suggested that the *tviA* 5' UTR harbors a *cis*-regulatory thermosensor.

We wanted to test whether the *tviA* 5' UTR structure controls *tviA* translation in response to fluctuating temperature. To do this, we generated a plasmid-based translational fusion of the *tviA* 5' UTR and a heat-stable β-galactosidase (i.e., *bgaB*) under transcriptional control of the arabinose-inducible P$_{BAD}$ promoter (Fig 2B) [37,42]. By growing *Escherichia coli* DH5α containing this fusion construct at 25, 37, and 42˚C and measuring reporter gene activity, we could determine if the putative *tviA* RNAT confers post-transcriptional regulation of β-galactosidase protein expression. We included the well-characterized *lcrF* fourU element from *Yersinia pseudotuberculosis* as a positive control and saw increasing β-galactosidase activity with increasing temperature, as expected (Fig 2C) [31]. The *tviA* 5' UTR fusion construct also showed a positive correlation between β-galactosidase activity and temperature, with more than 3-fold or 7-fold induction of BgaB activity from 25˚C to 37 or 42˚C, respectively (Fig 2C). This temperature-dependent synthesis of BgaB supports the hypothesis that the *tviA* 5' UTR contains an RNAT for control of virulence expression.

## Point mutations impair *tviA* thermosensor functionality

Previous studies demonstrated that targeted point mutations disrupt the RNAT structure, causing aberrant expression in response to temperature [31,35,39,41]. Substitution of noncanonical U-G pairs to more stable C-G pairs with stronger bonding between the fourU motif and the SD region creates a stabilized secondary structure in the RNAT, which cannot melt open to enable translation even at elevated temperatures. Conversely, disrupting G-C interactions between the SD and fourU sequences destabilizes the RNAT's overall structure, resulting in an open conformation and greater translation at low temperature [31,33,41]. Based on the proposed secondary structure, we designed point mutations predicted to either tighten or loosen the structure of the *tviA* thermosensor and introduced these mutations via site-directed mutagenesis (S3A and S3B Fig). The RNAT variants rep1 (T90C), rep2 (T92C), and rep3 (T90,92C; combination of rep1 and rep2) were predicted to repress the RNAT and prevent it from melting open with increasing temperature by increasing hydrogen bonds between the fourU element and the SD sequence. We also produced a derepressed variant (DEREP; T89,91G;C93G) predicted to weaken the fourU-SD interaction and enable a ribosome-accessible conformation at low temperature by eliminating two A-G and one stable C-G pairings. As described above, we fused the mutated *tviA* 5' UTRs to *bgaB* in pBAD2 and measured β-galactosidase activity at 25, 37, and 42˚C (Fig 2D) [42]. While repressive mutations rep1 and rep2 resulted in partial inhibition of *bgaB* expression (S3C Fig), we observed full repression for the rep3 variant (hereafter referred to as REP) even at 37 and 42˚C (Fig 2D). In contrast, the DEREP variant retained its temperature-responsiveness (Fig 2D). Furthermore, BgaB activity levels from the DEREP variant at 37 and 42˚C were higher than the wild-type (WT) *tviA* 5' UTR fusion construct, indicating a less stable RNAT structure. Importantly, neither the

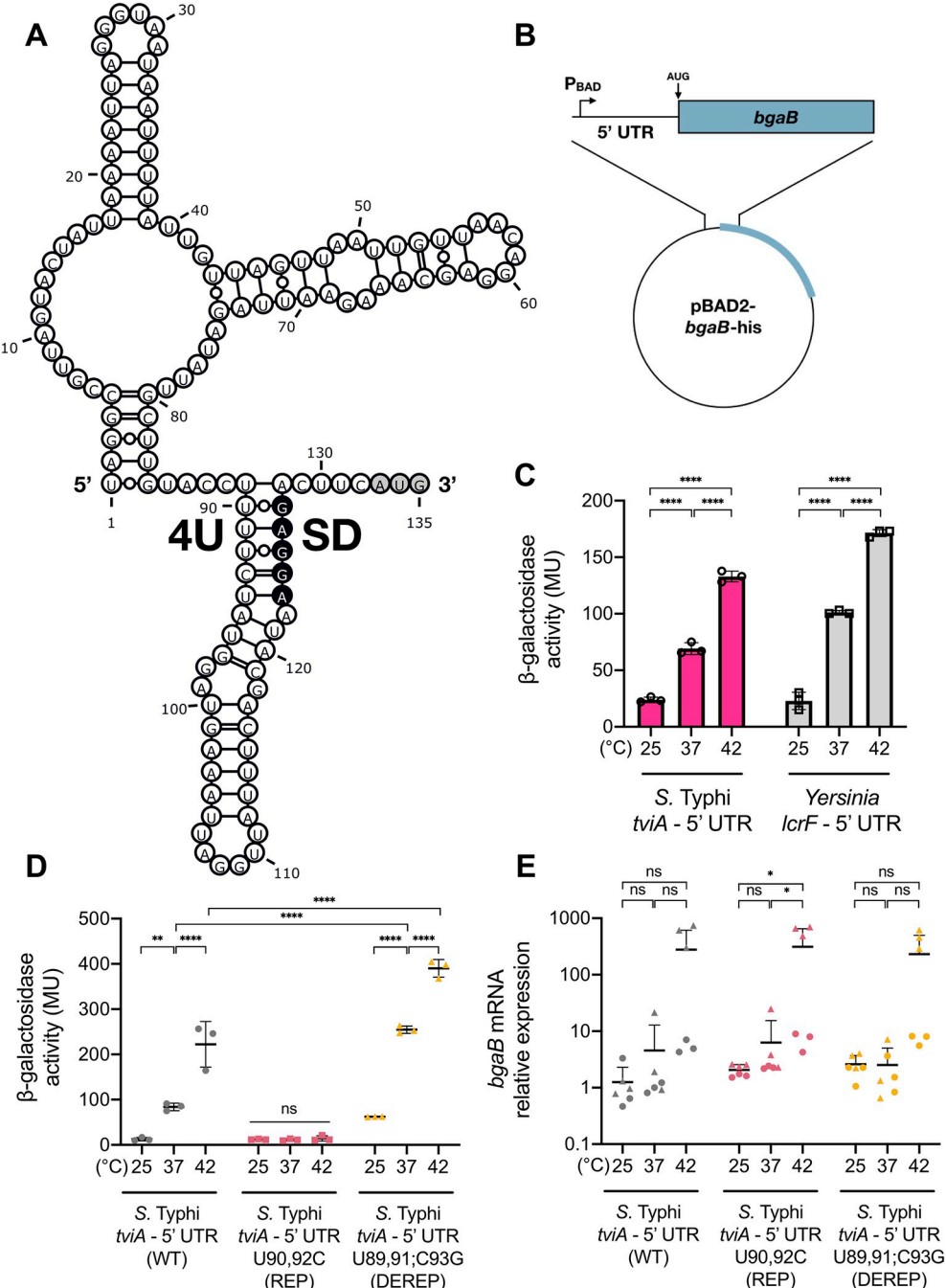

**Fig 2. The *tviA* 5' UTR mediates temperature-dependent translation.** A) Secondary structure prediction of the *tviA* 5' UTR (132 nt plus AUG codon; 135 nt total) using the Mfold program [36]. The start codon (AUG) is marked in gray. The Shine-Dalgarno (SD) and fourU RNA thermosensor (4U) sequences are indicated. B) Schematic of the RNA thermosensor-containing 5' UTR and heat-stable β-galactosidase (*bgaB*) fusion construct under transcriptional control of the arabinose inducible P_BAD promoter. C) Effect of temperature on *bgaB* translation from *S.* Typhi *tviA* and *Yersinia lcrF* 5' UTRs. D) Effect of altering base-pairing of the fourU region on temperature-dependent translation. Comparison of translation efficiency at different temperatures using the *S.* Typhi wildtype *tviA* 5' UTR (WT)-, the rep3 (T90,92C) *tviA* 5' UTR mutant (REP)-, and the derep4 (T89,91G;C93G) *tviA* 5' UTR mutant (DEREP)-*bgaB* fusion constructs. E) Relative expression levels of *bgaB* mRNA of the WT, REP, and DEREP *tviA* 5' UTR-*bgaB* fusion constructs following 30 minute temperature shifts. *bgaB* mRNA level normalized to 16S rRNA level for each sample. Data shown are representative of 3 (C and D) or are the combined results of 2 (E) independent experiments with triplicate samples for each condition. Symbol shape corresponds with different independent experiments. Data are represented as mean ± SD. NS, not significant. Statistical significance determined using one-way (C and E) or two-way (D) ANOVA with Tukey's correction. * $p < 0.05$, ** $p < 0.01$, **** $p < 0.0001$.

observed temperature-dependent increase in β-galactosidase activity nor the differences in activity between the *tviA* 5' UTR variants could be explained by *bgaB* transcript levels (Fig 2E). Although WT and DEREP showed higher *bgaB* transcript levels at 42°C compared to 25 and 37°C, this increase was not significant and this pattern was also seen with the REP variant, which showed negligible corresponding β-galactosidase activity levels at this temperature (Fig 2D and 2E). Additionally, the WT and DEREP variants did not show significant increases in *bgaB* transcript levels at 37°C compared to 25°C, even though we observed significantly more β-galactosidase activity at 37°C with these variants (Fig 2D and 2E). These results demonstrate that the secondary structure surrounding the SD region determines ribosome accessibility and that the thermodynamic interaction of the SD and fourU region enables precise temperature-dependent *tviA* translation.

## The *tviA* 5' UTR harbors a temperature-labile RNA structure

We next aimed to validate the proposed secondary structure of the *tviA* 5' UTR (Fig 2A) and to monitor its temperature-dependent unfolding using enzymatic structure probing of *in vitro*-transcribed, radiolabeled RNA. The properties of RNases T1 and T2, which cleave 3' of unpaired guanines and predominantly unpaired adenines, respectively, make them ideal for examining the melting behavior of the purine-rich SD sequence within RNAT structures [41,43,44]. Thus, we used these RNases to probe the *tviA* 5' UTR at 25, 37, and 42°C. Resolution of the cleavage products on an 8% polyacrylamide gel enabled us to assess temperature-dependent changes to the secondary structure conformation (Fig 3A).

For many residues, the cleavage pattern fit the predicted secondary structure, with nucleotides susceptible to RNase T1 and T2 cleavage at 25°C; however, some nucleotides that were predicted to be unpaired more likely adopted a double-stranded conformation based on their inability to be cleaved at 25°C (Figs 3A, 3B, and full gel in S4A). While only faint bands were visible at 25°C, the majority of the SD sequence was strongly prone to RNase T1 or T2 cleavage at 37 and 42°C, as indicated by increased band intensity (GGAG124-127; Figs 3A, S4A red highlight, S4B and S4C). This cleavage pattern indicates melting of the ribosome-binding region. Moreover, additional nucleotides (G101, AG117-118; S4A Fig brown highlight) were exclusively sensitive to enzymatic cleavage at 37 and 42°C, suggesting that the whole stem II is thermo-labile (Fig 3B). Apart from the WT *tviA* 5' UTR, we also probed the stabilized REP variant. Repressive mutations only affected the structure of stem loop II, resulting in a more stable structure (Figs 3A, 3B, S3A, S3B and S4A). Banding patterns that correspond to an open SD sequence were absent from RNA containing the repressive mutations (Figs 3A and S4A). These data indicate that the RNA structure is clamped in its "low temperature" conformation even at the elevated temperatures of 37 or 42°C. In contrast, the DEREP point mutations likely loosen the RNAT's structure, as the nucleotides of the SD region were sensitive to RNase T1/T2 cleavage at 25°C (Figs 3A and S4A). Furthermore, new band signals were detected at all three temperatures, corresponding to the anti-SD region (nucleotides 89 to 93; S4A Fig blue highlight) and include the mutated guanine residues.

Since the cleavage pattern of RNase T1/T2 (Figs 3A and S4A) deviated from the predicted secondary structure (Fig 3B), we used the information from enzymatic structure probing as hard constraints to guide folding of the *tviA* 5' UTR (Fig 3C and S1 Table). The guided structure prediction still consists of two hairpins, with the 3'-most harboring the SD sequence partially paired to the fourU motif. In contrast to the calculated minimal free energy structure (Fig 3B), the interior loop of hairpin II is enlarged (Fig 3C). We observed more drastic structural alterations within hairpin I with a multiloop comprising three individual hairpins. Taken together, the results clearly point to a temperature-sensitive structure within the *tviA* 5' UTR,

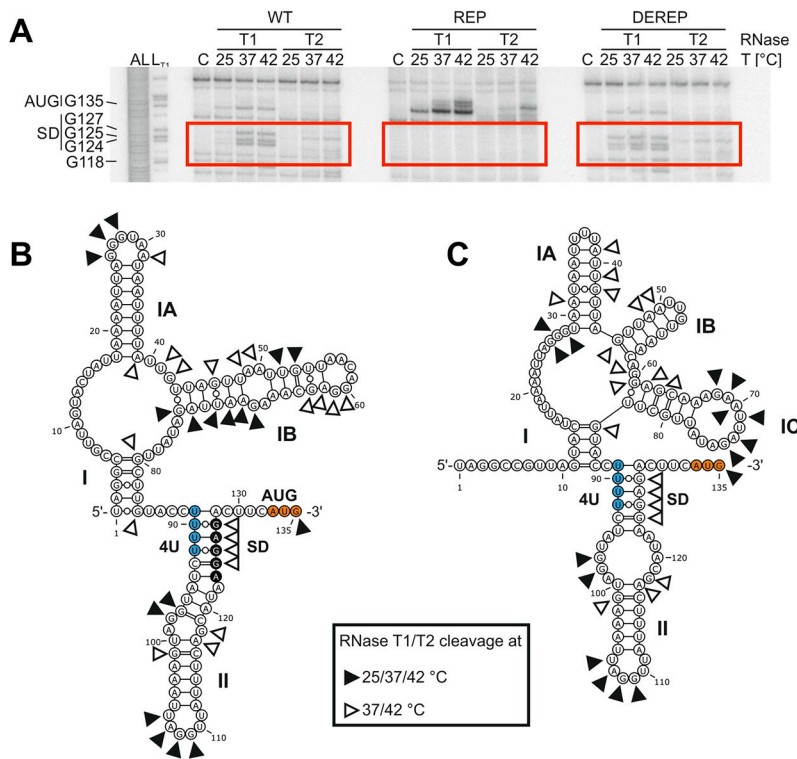

**Fig 3. The *tviA* 5' UTR fourU RNA thermosensor facilitates temperature-dependent access to the SD sequence.** A) 5' end-labeled *in vitro* transcribed RNA containing the wildtype *tviA* 5' UTR (WT), the rep3 (T90,92C) *tviA* 5' UTR mutant (REP), or the derep4 (T89,91G;C93G) *tviA* 5' UTR mutant (DEREP) was enzymatically probed with RNases T1 (cuts 3' of single-stranded guanines) and T2 (cuts 3' of single-stranded nucleotides with preference order: A > C > U > G) at 25, 37, and 42˚C. Fragmented RNA was separated on an 8% polyacrylamide gel, a portion of which is shown here. AL: alkaline ladder. L$_{T1}$: RNase T1 cleavage in sequence buffer at 37˚C. C: RNA treated with water instead of RNase at 42˚C. B) Secondary structure of the *tviA* 5' UTR predicted by Mfold [36]. Cleavage sites of RNase T1 and T2, the fourU RNA thermosensor (4U), the Shine-Dalgarno (SD) sequence, and the start codon (AUG) are indicated. C) Secondary structure of the *tviA* 5' UTR predicted by Mfold using cleavage information as constraints for folding. Secondary structures were visualized using VARNA applet 3.93 [45]. Data shown in A are representative of 4 independent experiments. See also S4 Fig.

which enhances accessibility to the ribosome binding site under virulence-relevant conditions. Notably, point mutations within the predicted binding sites manipulate the melting behavior of the RNAT.

## Melting of the *tviA* RNA thermosensor favors ribosome binding at 37˚C

The zipper-like melting process of RNATs enables increased ribosome-binding to the SD sequence at elevated temperatures [38]. To address whether the *tviA* RNAT allows temperature-dependent ribosome-binding, we performed toeprinting experiments [46]. We generated *in vitro* transcribed RNA fragments consisting of the *tviA* RNAT as well as 60 nucleotides of the coding region and performed reverse transcription in the presence or absence of the 30S ribosomal subunit at 25, 37, and 42˚C. After primer extension, we separated the reverse transcription products on an 8% denaturing polyacrylamide gel. If ribosomes do not bind the RNA fragment, polymerase extension will occur unimpeded, resulting in a full-length reverse transcription product (Fig 4A, top of gel). However, ribosomes that bind the RNA fragment prevent polymerase extension, generating shorter products and a "toeprint." We observed toeprinting signals at position +15 to +17 relative to the translation start codon (AUG) only in the

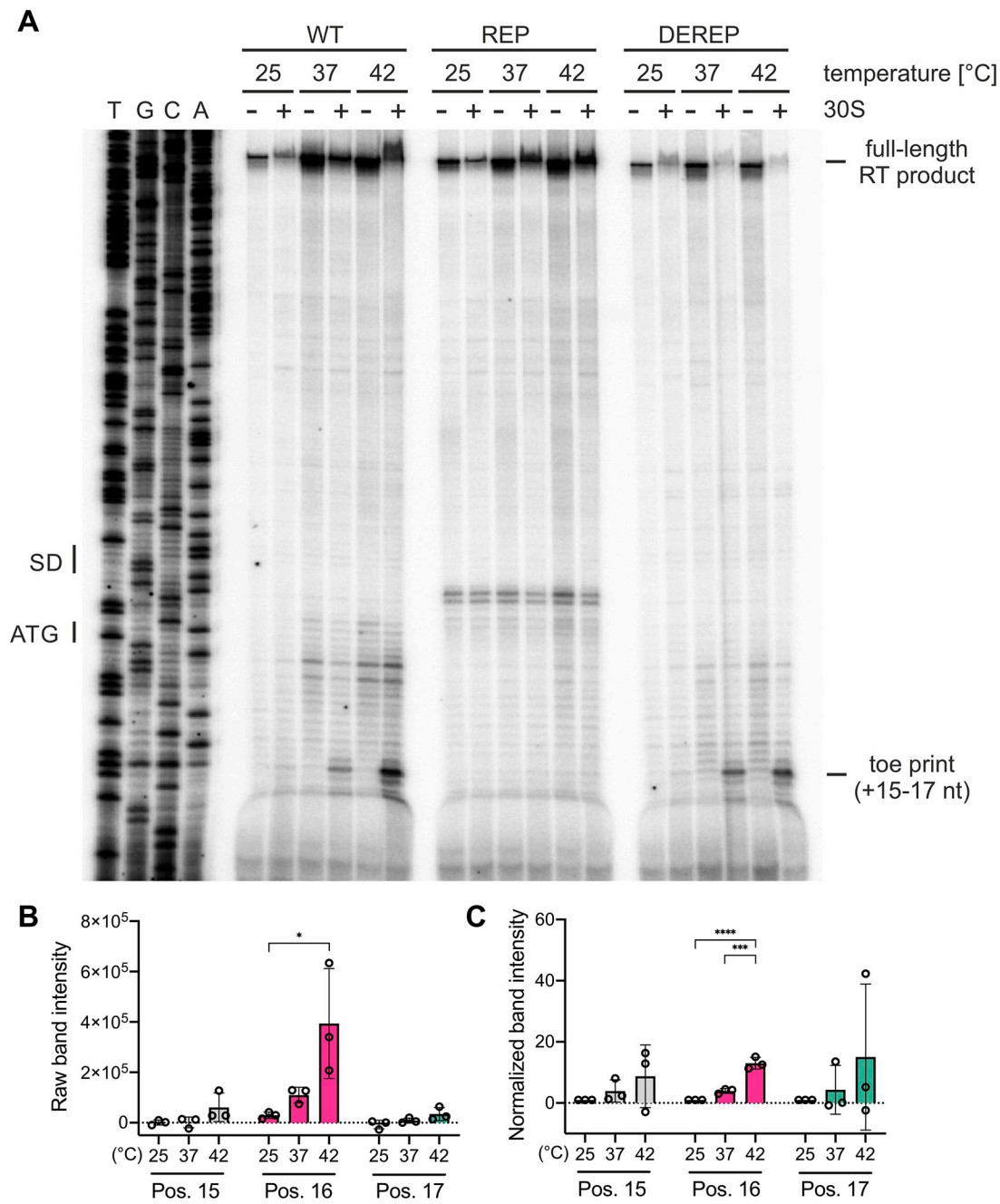

**Fig 4. Temperature-dependent binding of ribosomes to the *tviA* 5' UTR.** A) Toeprint analysis was performed with *in vitro* transcribed RNA containing the wildtype *tviA* 5' UTR (WT), the rep3 (T90,92C) *tviA* 5' UTR mutant (REP), or the derep4 (T89,91;C93G) *tviA* 5' UTR mutant (DEREP) at 25, 37, and 42˚C, as described in Materials and Methods. TGCA indicates the corresponding DNA sequencing reactions, and the positions of the SD sequence and the ATG start codon are indicated. Reverse transcription reactions were performed without (-) and with (+) the 30S ribosomal subunit. The signals corresponding to the full-length transcript and the toeprint corresponding to the position +15 to +17 from the start codon are indicated. B and C) For toeprint positions +15, +16, and +17, the raw band intensity (B) and normalized band intensity (C) was calculated at each temperature tested for the WT 5' UTR. For normalization, the band intensity of each position at each temperature was normalized to the average band intensity of that position at 25˚C. Data shown are representative of 3 independent experiments. Statistical significance determined using one-way ANOVA with Tukey's correction. * p < 0.05, *** p < 0.001, **** p < 0.0001.

presence of ribosomal subunits (Fig 4A). Toeprint signal intensity also increased with increasing temperature (Fig 4A, 4B and 4C), indicating improved ribosomal binding at higher temperatures. We did not detect a toeprinting signal with the REP variant of the *tviA* thermometer (Fig 4A), which is consistent with the absence of RNA melting *in vitro* and translation initiation *in vivo*. Moreover, a faint signal around the SD sequence was observed similar to previously described repressed RNAT structures [31,44,47,48]. This pattern may be explained by the stabilized secondary structure of the repressed variant causing the reverse transcriptase to prematurely stop or drop off, generating a toeprint signal despite a lack of ribosome binding [49]. There was no difference in band signal intensity between the WT and DEREP RNATs (Fig 4A). Nevertheless, these results demonstrate ribosomal subunit binding in response to elevated temperature. Furthermore, this demonstrates that increased temperature is sufficient for ribosomal binding within the RNAT independent of any additional bacterial factors.

## The *tviA* RNAT regulates *S.* Typhi virulence factor expression in response to temperature

To determine the biological relevance of the *tviA* RNAT in its native setting, we introduced the REP (*tviA*-REP) and DEREP (*tviA*-DEREP) mutations into the 5' UTR of *tviA* in *S.* Typhi. As we saw previously (Fig 1), Vi capsule expression positively correlated with increasing temperature in WT Ty2 *S.* Typhi, and, intriguingly, WT *S.* Typhi expressed even more Vi capsule at 42°C than at 37°C (Fig 5A). RNAT repressing mutations abrogated the temperature-dependent Vi capsule expression phenotype in the *S.* Typhi *tviA*-REP mutant, resulting in Vi capsule production levels similar to the levels in a *S.* Typhi Δ*tviA* mutant (Fig 5A). In contrast, the *S.* Typhi *tviA*-DEREP mutant preserved temperature-dependent increases in Vi capsule expression and showed significantly higher levels of Vi capsule at both 23°C and 37°C compared to WT (Fig 5A). Importantly, the temperature-dependent increases in Vi production in WT or *tviA*-DEREP *S.* Typhi were not due to temperature-dependent increases in *tviA* mRNA transcript levels (Fig 5B). We observed lower levels of *tviA* mRNA in the *S.* Typhi *tviA*-REP mutant at all temperatures tested. This is most likely explained by transcript instability due to a lack of ribosome binding [37]. Together, these data suggest that the 5' UTR of *tviA* in *S.* Typhi contains a functional RNAT that regulates Vi capsule expression in a temperature-dependent manner.

Since TviA suppresses expression of flagellin, we wanted to elucidate the role of the *tviA* RNAT in modulating *S.* Typhi motility [15]. We inoculated low agar concentration plates containing salt levels that induce (0 mM NaCl) or suppress (150 mM NaCl) *tviA* transcription with *S.* Typhi strains, grew them at 37°C, and measured halo size as an indicator of motility (Fig 5C and 5D). In low salt conditions, the halo size produced by WT *S.* Typhi was significantly smaller than the halo produced in high salt conditions, indicating production of TviA and suppression of flagellin, as expected. In addition, the halo sizes produced by the *S.* Typhi Δ*fliC* mutant were small, regardless of salt concentration, consistent with the presence of flagellin being required for motility. In contrast, the halo sizes produced by the *S.* Typhi Δ*tviA* mutant were large, indicating that the inability to suppress flagellin expression leads to motility regardless of salt concentration (Fig 5C and 5D). Importantly, the *S.* Typhi *tviA*-REP mutant produced large halos when grown under both salt concentrations, consistent with the absence of TviA in this *S.* Typhi strain. In contrast, the *S.* Typhi *tviA*-DEREP mutant produced halos that were smaller than the WT *S.* Typhi strain under low salt conditions. Collectively, our results indicate that a functional *tviA* RNAT is important for *S.* Typhi motility via modulation of flagellin.

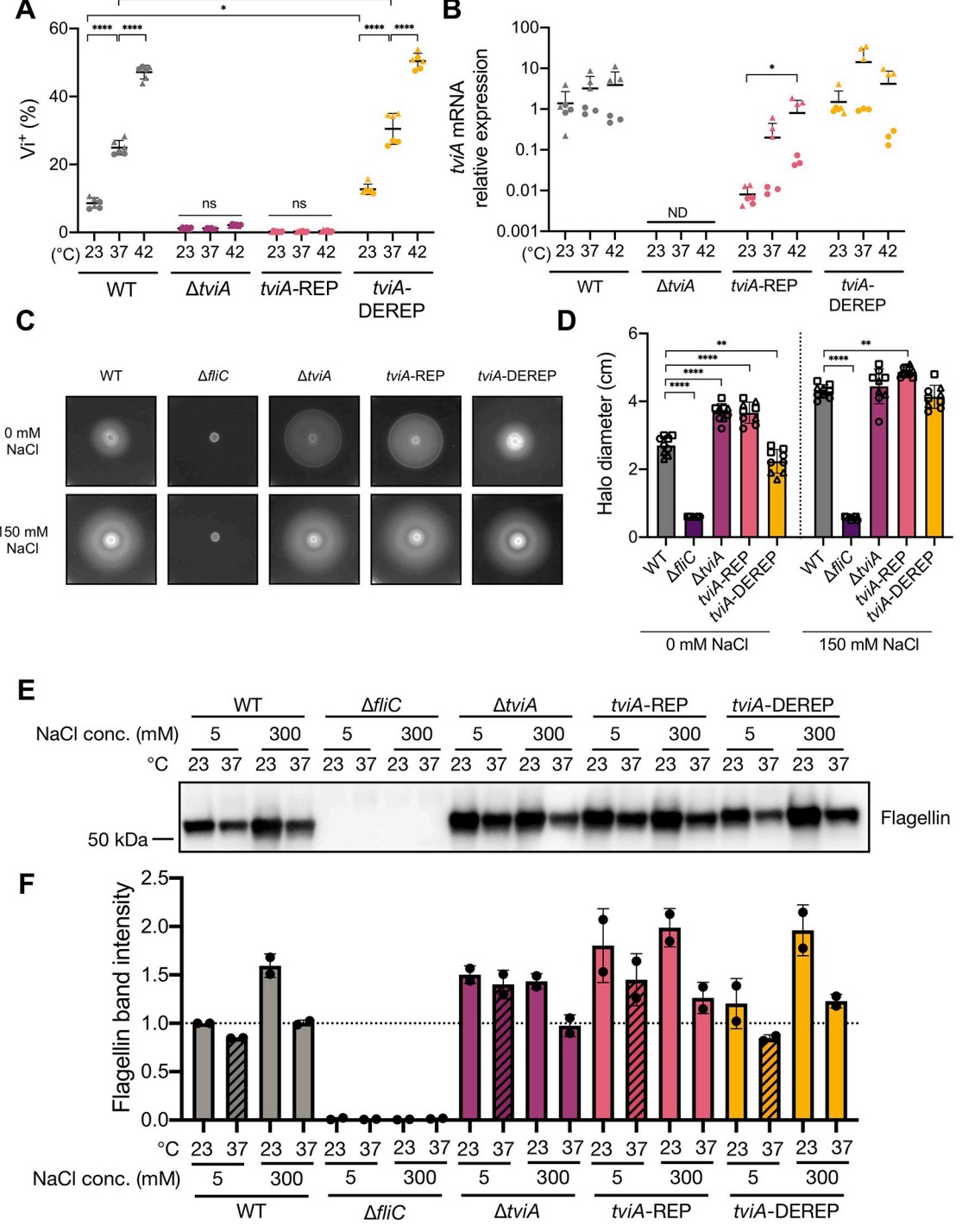

**Fig 5. The *tviA* 5' UTR fourU RNA thermosensor regulates virulence factor expression in a temperature-dependent manner.**
A) Quantitation of Vi capsule expression by flow cytometry of wild-type (WT), Δ*tviA*, *tviA*-REP (T90,92C), or *tviA*-DEREP (T89,91G;C93G) *S*. Typhi at ambient temperature (23°C) or following a shift to human body temperature (37°C) or fever-like temperature (42°C) for 30 minutes. B) Relative expression levels of *tviA* mRNA of WT, Δ*tviA*, *tviA*-REP, or *tviA*-DEREP *S*. Typhi at ambient temperature (23°C) or following a shift to human body temperature (37°C) or fever-like temperature (42°C) for 30 minutes. *tviA* mRNA levels normalized to 16S rRNA levels for each sample. C) Representative images of WT, Δ*fliC*, Δ*tviA*, *tviA*-REP, or *tviA*-DEREP *S*. Typhi grown on motility assay plates (0.3% agar) containing 0 mM or 150 mM NaCl at 37°C for 24 hours. D) Measurement of halo diameter (in cm) of WT, Δ*fliC*, Δ*tviA*, *tviA*-REP, or *tviA*-DEREP *S*. Typhi grown on motility assay plates as described in 5C. E) Western blot for flagellin from whole cell lysates of WT, Δ*fliC*, Δ*tviA*, *tviA*-REP, or *tviA*-DEREP *S*. Typhi

grown for 24 hours statically under the following conditions: low salt media (5 mM NaCl LB broth) at 23˚C, low salt media at 37˚C, high salt media (300 mM NaCl LB broth) at 23˚C, or high salt media at 37˚C. F) Quantitation of flagellin from Western blots. Flagellin band intensities were normalized to total protein band intensities for each sample. These values were then further normalized against the "WT low salt 23˚C" sample for the purposes of plotting the data. The quantitation results from two independent Western blots are shown. Data shown are combined from 2 (A, B, and F) or 3 (D) independent experiments with triplicate samples for each condition or representative of 2 (E) or 3 (C) independent experiments. Symbol shape corresponds with different independent experiments. Data are represented as mean ± SD. ND, not detected. NS, not significant. Significance calculated using one-way (B and D) or two-way (A) ANOVA with Tukey's correction. * p < 0.05, ** p < 0.01, **** p < 0.0001.

To further explore the role of the *tviA* RNAT in modulating flagellin, we investigated flagellin production under growth conditions that combined the effects of both temperature and osmolarity. We probed for flagellin via Western blot using whole cell lysates from *S.* Typhi strains grown statically under the following conditions: low salt media (5 mM NaCl LB broth) at 23˚C, low salt media at 37˚C, high salt media (300 mM NaCl LB broth) at 23˚C, or high salt media at 37˚C. As expected, the Δ*fliC* mutant of *S.* Typhi did not produce flagellin under any of the tested conditions (Figs 5E, 5F, S5A and S5B). WT *S.* Typhi produced low levels of flagellin in low salt media at 37˚C, which was expected since low salt induces *tviA* transcription and high temperature melts open the *tviA* RNAT and allows for maximal translation, leading to high levels of TviA and maximal flagellin suppression (Figs 5E, 5F, S5A and S5B). In contrast, WT *S.* Typhi exhibited the highest levels of flagellin production in high salt media at 23˚C (Figs 5E, 5F, S5A and S5B). This result was also expected, since these conditions should maximally suppress TviA production, enabling flagellin expression. The remaining two conditions resulted in intermediate levels of flagellin production in WT *S.* Typhi (Figs 5E, 5F, S5A and S5B). This was expected since the low salt low temperature condition should allow for *tviA* transcription but not translation, and the high salt high temperature condition should allow for translation from any *tviA* mRNA present but inhibit synthesis of additional transcripts. Similar to WT *S.* Typhi, the *tviA*-DEREP mutant produced the lowest levels of flagellin in low salt media at 37˚C and the highest levels of flagellin in high salt media at 23˚C (Figs 5E, 5F, S5A and S5B). These results were expected, since the *tviA*-DEREP *S.* Typhi mutant still contains a functional *tviA* RNAT. It appeared that the *tviA*-DEREP mutant may have slightly lower levels of flagellin production under conditions of low salt and high temperature compared to WT *S.* Typhi (Figs 5E, 5F, S5A and S5B; compare patterned bars in Figs 5F and S5B). This result would fit with the derepressing mutations opening up the *tviA* RNAT structure more than in WT *S.* Typhi, enabling greater production of TviA and more suppression of flagellin expression. However, we cannot conclude based on these data whether the observed difference is significant. Importantly, the *tviA*-REP mutant produced more flagellin than WT *S.* Typhi when grown in low salt media at 37˚C, and the amount of flagellin produced was comparable to the level produced by the Δ*tviA* mutant under these conditions (Figs 5E, 5F, S5A and S5B; compare patterned bars in Figs 5F and S5B). This is consistent with the repressing mutations preventing TviA production, leading to an inability to suppress flagellin. Across the tested conditions, both the Δ*tviA* and *tviA*-REP mutants appeared to have greater flagellin production than WT *S.* Typhi, which is consistent with an absence of TviA production. However, we observed some differences in the level of flagellin production in these mutants between the tested conditions. It is possible that another regulatory element involved in flagellin expression may be sensitive to either osmolarity and/or temperature and modulate flagellin expression in the absence of TviA. However, our results suggest that TviA plays a dominant role in modulating flagellin expression. Additionally, our results support the conclusion that a functional *tviA* RNAT is critical for regulating flagellin expression in response to human host body temperature.

### *S.* Typhi evasion of innate immune signaling in intestinal epithelial cells requires the *tviA* RNAT

Given that flagellin is a major PAMP that is recognized by the human innate immune system and its expression is modulated by TviA, we wanted to determine the role the *tviA* RNAT plays in innate immune responses [15]. Before *S.* Typhi establishes a systemic infection, it must first translocate the intestinal epithelial cell (IEC) barrier. IECs serve as the first line of defense between the gut lumen and systemic tissues, and they basolaterally express TLRs to detect the presence of invasive pathogens [19,50,51]. Previous work has demonstrated that *S.* Typhi uses TviA to modulate its virulence factor expression to evade detection by TLRs and prevent intestinal inflammation in order to establish successful systemic infection [9,11,18,52]. Thus, we hypothesized that a functional *tviA* RNAT would be critically important for evasion of TLR-dependent signaling in IECs. To test this, we infected the human colonic epithelial cell line HT-29 with *S.* Typhi mutants grown under non-invasive conditions (i.e., low salt) and measured secretion of the pro-inflammatory cytokine IL-8 as an indication of TLR-dependent signaling at 6 and 18 hours post-infection (p.i.) (Fig 6A and 6B). As expected, the *S.* Typhi Δ*tviA* mutant strain induced significantly higher levels of IL-8 secretion at both 6 and 18 hours p.i. compared to WT *S.* Typhi. (Fig 6A and 6B). Importantly, the *S.* Typhi *tviA*-REP mutant induced higher levels of IL-8 secretion at 6 and 18 hours p.i. compared to WT *S.* Typhi, similar to the TviA-deficient *S.* Typhi strain (Fig 6A and 6B). To test whether the increased levels of cytokine secretion induced by the *S.* Typhi *tviA*-REP mutant strain were dependent on flagellin, we deleted the *fliC* gene in this strain. Indeed, the *S.* Typhi *tviA*-REP Δ*fliC* mutant strain induced significantly lower levels of IL-8 secretion at 6 and 18 hours p.i. (Fig 6A and 6B) compared to the *S.* Typhi *tviA*-REP mutant strain, demonstrating that the inability of the *S.* Typhi *tviA*-REP mutant to suppress flagellin expression leads to greater innate immune signaling. We also observed that WT and the *tviA*-DEREP *S.* Typhi strains induced secretion of IL-8 at levels that were the same as the *S.* Typhi Δ*fliC* mutant (Fig 6A and 6B). This was expected since low salt growth conditions were used, which should induce maximal suppression of flagellin

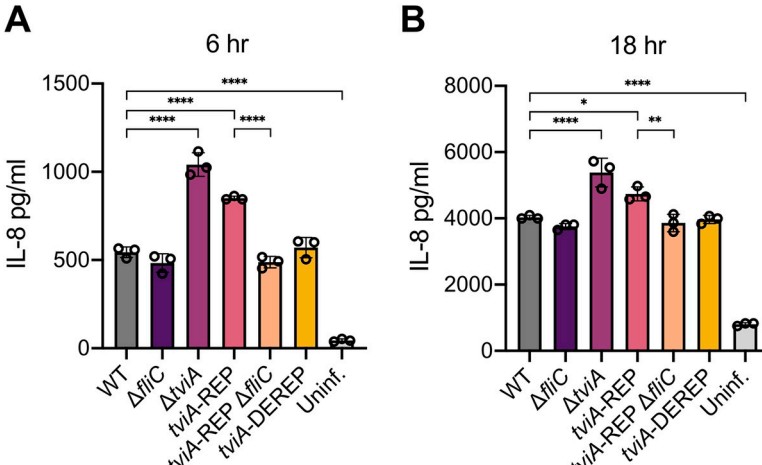

**Fig 6. The *tviA* 5' UTR fourU RNA thermosensor is required for *S.* Typhi evasion of innate immune signaling in human intestinal epithelial cells.** A and B) TLR-dependent signaling as measured by IL-8 secretion from the human intestinal epithelial cell line HT-29 following infection for 6 (A) or 18 (B) hours with non-invasive WT, Δ*fliC*, Δ*tviA*, *tviA*-REP, *tviA*-REP Δ*fliC*, or *tviA*-DEREP *S.* Typhi at MOI = 25 or left uninfected. Data shown are representative of 2 (B) or 3 (A) independent experiments with triplicate samples for each condition. Data are represented as mean ± SD. Significance calculated using one-way ANOVA with Tukey's correction. ** $p < 0.01$, *** $p < 0.001$, **** $p < 0.0001$.

expression in both WT and *tviA*-DEREP *S.* Typhi. Taken together, our results indicate that a functional *tviA* RNAT is important for *S.* Typhi to modulate flagellin expression in order to evade innate immune signaling upon infection of IECs.

### *tviA* RNAT regulation of virulence factors is critical for evasion of inflammasome activation

Cytosolic flagellin is a potent activator of the NAIP/NLRC4 inflammasome, leading to caspase-1 activation and cleavage, pyroptotic cell death, and release of the pro-inflammatory cytokines IL-1β and IL-18 [21–23,26,29]. Previous work has shown that TviA is important for reducing flagellin-mediated inflammasome activation by *S.* Typhi [27]. Since the *S.* Typhi *tviA*-REP mutant cannot suppress flagellin expression, we hypothesized that infection with this strain would lead to greater inflammasome activation compared to WT *S.* Typhi. To test this, we infected LPS-primed human macrophage-like THP1 cells with *S.* Typhi mutants and assayed for pyroptotic cell death with lactate dehydrogenase (LDH) release and IL-1β release at 2 hours post-infection as well as cell death kinetics using SYTOX Green staining over 10 hours of infection (Fig 7A, 7B and 7C). As predicted, the *S.* Typhi *tviA*-REP mutant induced significantly higher levels of macrophage death (Fig 7A and 7C) and IL-1β release (Fig 7B) compared to WT *S.* Typhi. In addition, the *tviA*-REP mutant and the TviA-deficient *S.* Typhi strains induced similar increases in the levels of cell death and IL-1β release compared to WT *S.* Typhi (Fig 7A, 7B and 7C), consistent with the role of the *tviA* RNAT controlling expression of TviA-regulated genes, such as flagellin. To test whether the increased inflammasome activation in macrophages infected with the *S.* Typhi *tviA*-REP mutant was dependent on flagellin, we infected macrophages with the *S.* Typhi *tviA*-REP Δ*fliC* mutant strain. The levels of macrophage death (Fig 7A and 7C) and IL-1β release (Fig 7B) from macrophages infected with the *S.* Typhi *tviA*-REP mutant deficient for flagellin were the same as the very low levels that were induced by the WT *S.* Typhi strain (Fig 7A, 7B and 7C). Although macrophages infected with WT and *tviA*-DEREP *S.* Typhi strains induced similar levels of macrophage death, the *S.* Typhi *tviA*-DEREP mutant strain caused significantly less IL-1β release (Fig 7B), suggesting the derepressed *tviA* RNAT enables *S.* Typhi to evade inflammasome activation to a greater extent than the wild-type RNAT. Together, these data suggest that a functional *tviA* RNAT is critical for *S.* Typhi to suppress flagellin expression upon entering the host cell, preventing inflammasome activation and preserving *S.* Typhi's replicative niche.

To understand better the role of the *tviA* RNAT in human infection, we infected primary human monocyte-derived macrophages (hMDMs) from two different donors with *S.* Typhi mutants and assessed inflammasome activation. *S.* Typhi Δ*tviA* and *tviA*-REP mutant strains caused significantly higher levels of cell death (LDH release and SYTOX Green staining) and IL-1β release in hMDMs than WT *S.* Typhi, and this was due to excessive flagellin expression, since deletion of *fliC* from the *S.* Typhi *tviA*-REP mutant strain reduced macrophage death and IL-1β levels to those of WT-infected hMDMs (Fig 7D, 7E, 7F, 7G and 7H). Additionally, the *S.* Typhi Δ*tviA* and *tviA*-REP mutant strains induced higher levels of caspase-1, IL-1β, and gasdermin D cleavage as assessed by Western blot compared to the WT *S.* Typhi strain, indicating greater inflammasome activation upon infection with these mutants (Figs 7I and S6). This increase in inflammasome activation is most likely due to detection of flagellin by the NAIP/NLRC4 inflammasome, since the *S.* Typhi *tviA*-REP Δ*fliC* mutant induced less caspase-1, IL-1β, and gasdermin D cleavage than the *S.* Typhi *tviA*-REP mutant. These results suggest that the *tviA* RNAT is critical for evasion of inflammasome activation during infection of primary human cells.

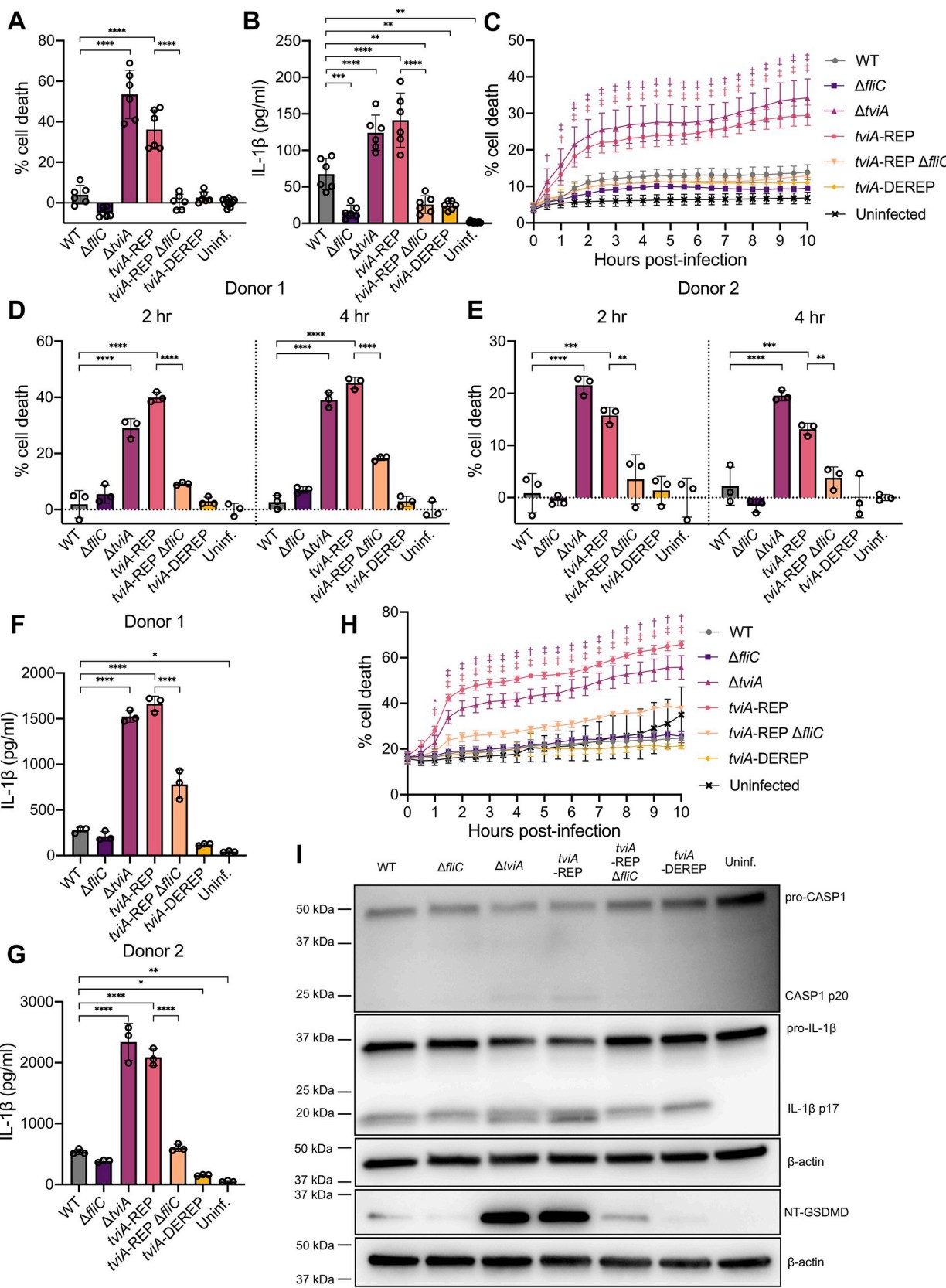

**Fig 7. The *tviA* 5' UTR fourU RNA thermosensor enables *S.* Typhi to evade inflammasome activation during infection of human macrophages.** A) Pyroptotic cell death as measured by lactate dehydrogenase release from the human macrophage-like cell line THP1 primed overnight with 100 ng/ml LPS and following infection for 2 hours with WT, Δ*fliC*, Δ*tviA*, *tviA*-REP, *tviA*-REP Δ*fliC*, or *tviA*-DEREP S. Typhi at MOI = 10 or left uninfected. B) Corresponding release of IL-1β to infection described in 7A. C) Kinetics of THP1 cell death as measured by SYTOX Green positive staining on an IncuCyte S3 after overnight priming with 100 ng/ml LPS and infection with WT, Δ*fliC*, Δ*tviA*, *tviA*-REP, *tviA*-REP Δ*fliC*, or *tviA*-DEREP *S.* Typhi at MOI = 10 or left uninfected. Symbols indicate statistical comparison of Δ*tviA* or *tviA*-REP to WT at each time point. D and E) Pyroptotic cell death as measured by lactate dehydrogenase release from primary human monocyte-derived macrophages (hMDMs) from two different donors (Donor 1 D and Donor 2 E) primed overnight with 100 ng/ml LPS and following infection for 2 or 4 hours with WT, Δ*fliC*, Δ*tviA*, *tviA*-REP, *tviA*-REP Δ*fliC*, or *tviA*-DEREP S. Typhi at MOI = 10 or left uninfected. F and G) Corresponding release of IL-1β to 2 hour post-infection time point described in 7D-E. H) Kinetics of hMDM cell death as measured by SYTOX Green positive staining on an IncuCyte S3 after overnight priming with 100 ng/ml LPS and infection with WT, Δ*fliC*, Δ*tviA*, *tviA*-REP, *tviA*-REP Δ*fliC*, or *tviA*-DEREP *S.* Typhi at MOI = 10 or left uninfected. Asterisks and symbols indicate statistical comparison of Δ*tviA* or *tviA*-REP to WT at each time point. I) Western blot of caspase-1 (CASP1), IL-1β, and cleaved N-terminal gasdermin D (NT-GSDMD) in cell lysate plus supernatant of hMDM cells primed overnight with 100 ng/ml LPS and following infection for 2 hours with WT, Δ*fliC*, Δ*tviA*, *tviA*-REP, *tviA*-REP Δ*fliC*, or *tviA*-DEREP *S.* Typhi at MOI = 25 or left uninfected. Data shown are pooled from 2 (A, B, and C) independent experiments, are individual experiments with hMDMs from two different primary human blood donors (D, E, F, and G), or representative of 2 (H and I) independent experiments performed with hMDMs derived from two different primary human blood donors. Triplicate samples for each condition were used in the experiments in A, B, C, D, E, F, G, and H. Data are represented as mean ± SD. Significance calculated using one-way ANOVA with Tukey's correction. * $p < 0.05$, ** $p < 0.01$, *** $p < 0.001$, **** $p < 0.0001$. In C and H, † indicates *** significance and ‡ indicates **** significance.

## Conservation of the fourU RNAT in Vi capsule-producing, enteric fever-causing human pathogens

Other *Salmonella enterica* serovars have acquired the *viaB* locus and express Vi capsule. These include the human-specific pathogen *S.* Paratyphi C, which also causes enteric fever, and some isolates of *S.* Dublin, which causes an enteric fever-like disease in humans with high fever and bacteremia but little gastrointestinal disease [53–55]. Interestingly, Vi capsule expression is not constrained to *Salmonella* spp. Both the opportunistic human pathogens *Citrobacter freundii* and *Bordetella petrii* as well as soil bacteria in the genus *Achromobacter* have homologs of *viaB* genes and produce Vi capsule [56]. Unlike Vi capsule-producing *Salmonella* spp., little is known about regulation of capsule expression in *B. petrii* and *Achromobacter* spp., but the genomes of these organisms do not contain a *tviA* homolog [56]. Although *C. freundii* regulates its Vi capsule expression via an insertion sequence 1-like element that inserts into a recombinational hotspot in the *viaB* locus, it does have a *tviA* homolog called *vipR* [57]. Alignment of the *tviA* 5' UTRs from *Salmonella* spp. as well as the *C. freundii vipR* 5' UTR sequence revealed conservation of the fourU RNAT motif within all the *tviA* sequences, whereas *C. freundii vipR* lacks the fourU RNAT motif (S7 Fig). Interestingly, the *S.* Paratyphi C and *S.* Dublin *tviA* 5' UTR sequences lack the bulging adenine immediately 5' to the SD region in the *S.* Typhi *tviA* 5' UTR, which may stabilize the *tviA* 5' UTR structure, requiring higher temperatures to melt the RNAT and enable translation in these serovars (Figs 2A and S7). However, without experimentally testing the *S.* Paratyphi C and *S.* Dublin *tviA* 5' UTRs against the *S.* Typhi *tviA* 5' UTR, the effect of the bulging adenine on RNAT functionality cannot be definitively concluded. Additionally, although it is possible that the *vipR* 5' UTR still functions as an RNAT despite the lack of the fourU motif, the results of this alignment suggest that regulation of virulence factor expression in response to temperature is a key aspect of *Salmonella* spp. that express Vi capsule and cause enteric fever in humans.

## Discussion

Our results demonstrate that *S.* Typhi modulates virulence factor expression through an RNAT-mediated temperature sensing mechanism. At the human core body temperature of 37˚C, *S.* Typhi synthesizes more Vi capsule than at lower temperatures. Simultaneously, *S.* Typhi suppresses flagellin expression at 37˚C. This coordinated regulation of virulence factor expression in response to temperature is controlled by the transcriptional regulator TviA. Our

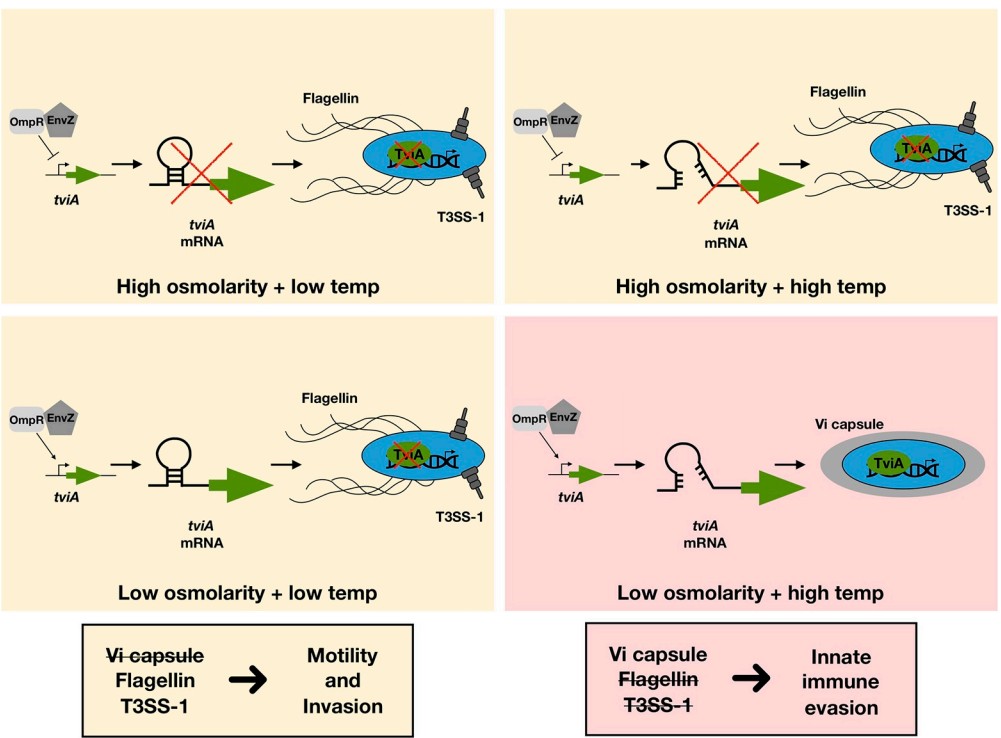

**Fig 8. Production of TviA is a two-checkpoint system.** Although EnvZ/OmpR induce *tviA* transcription under low osmolarity conditions, production of the TviA protein only occurs in low salt environments with high temperature due to thermoregulation of *tviA* translation by the fourU RNAT. This fine-tuning of TviA expression enables *S*. Typhi to precisely regulate virulence factor expression (i.e., Vi capsule, flagellin, T3SS-1) for optimal evasion of host innate immune responses.

work shows that one mechanism dictating temperature-dependent control of virulence factor expression is mediated by an RNAT in the 5' UTR of *tviA* that exerts post-transcriptional control of TviA protein production. Using *in vitro* methods and point mutations in the *tviA* 5' UTR that either stabilize or destabilize the RNAT structure, we showed that the RNAT melts open at higher temperatures to enable ribosome access to the SD region and subsequent translation of *tviA*. These same point mutations regulated temperature-dependent virulence factor expression in *S*. Typhi Ty2 and impacted innate immune responses during *in vitro* infection. Our results are consistent with previous studies demonstrating that TviA-mediated regulation of flagellin is important for extracellular *S*. Typhi to evade TLR5 stimulation of IECs and intracellular *S*. Typhi to evade NAIP/NLRC4 inflammasome activation in macrophages [27]. Thus, *tviA* RNAT-mediated regulation of virulence factor expression enables *S*. Typhi to rapidly alter its protein expression profile in response to temperature, allowing it to evade detection by the host innate immune response.

Previous work has demonstrated the importance of environmental osmolarity in TviA production [15]. Our work suggests regulation of TviA production integrates multiple environmental stimuli and involves at least two checkpoints. Osmolarity serves as the initial checkpoint, dictating whether or not *tviA* is transcribed, and temperature serves as the second checkpoint, post-transcriptionally controlling *tviA* translation (Fig 8). This fine-tuning suggests that *S*. Typhi has evolved to precisely coordinate its protein expression profile for distinct tissue locations during infection of the human host. Although *S*. Typhi suddenly experiences an increase in temperature upon ingestion, the high osmolarity of the intestinal lumen

prevents transcriptional induction of *tviA*. Upon trafficking into the intestinal epithelium and establishing intracellular infection, *S.* Typhi experiences both low salt and high temperature conditions, permitting TviA synthesis. These transcriptional and translational checkpoints ensure that TviA production, and expression of the virulence factors it subsequently modulates, occurs at the optimal spatiotemporal moment during establishment of infection in a new host. Additionally, this two-checkpoint system could benefit *S.* Typhi outside of the human host. In the external environment, *S.* Typhi may experience low osmolarity conditions and induce *tviA* transcription, but the ambient temperature may not be permissive for translation to occur. This could prevent *S.* Typhi from expending precious energetic resources modulating virulence factor expression in a context where it serves no fitness benefit.

Typhoid fever derives its name from the characteristic high fever patients exhibit upon acute infection with *S.* Typhi [4,5]. Fever is an ancient, common innate immune response that helps the host defend against infection by accelerating immune functions and restricting pathogen growth [58,59]. The titratability of Vi capsule expression in response to increasing temperature—and in particular to the high temperature of 42˚C—suggests that instead of being blunted by fever, *S.* Typhi may exploit this host response to further modulate virulence factor expression to evade innate immune detection. Unlike *S.* Typhimurium, which not only causes high levels of intestinal inflammation but thrives on it, *S.* Typhi is frequently described as a "stealth" pathogen because of its ability to induce little intestinal inflammation by evading immune detection, enabling it to spread systemically in the host [5]. Utilization of an RNAT to induce Vi capsule and suppress PAMP expression suggests that *S.* Typhi has evolved to take advantage of the fever response it induces upon systemic infection to further subvert detection and persist within the host.

RNATs have been identified in several critical human bacterial pathogens [2]. Interestingly, many of the characterized RNATs occur in the 5' UTR upstream of genes encoding transcriptional regulators. These include *prfA* in *Listeria monocytogenes*, *toxT* in *Vibrio cholerae*, *lcrF* in *Yersinia pseudotuberculosis*, and *rhlR* and *ptxS* in *Pseudomonas aeruginosa* [31–35]. Our work is the first to identify an RNAT in a critical virulence factor transcriptional regulator in the human-specific pathogen *S.* Typhi. The evolution of RNAT-mediated post-transcriptional control of transcription factor expression across these bacterial pathogens suggests that this temperature-response mechanism is a rapid and efficient method to adapt to the host environment. Additionally, these transcriptional regulators control the expression of major virulence factors for each of these pathogens. Thus, RNATs specifically enable quick and efficient modulation of virulence factor expression, allowing bacterial pathogens to rapidly subvert and defend against host immune responses. It is striking that a few base pair alterations can completely disrupt the RNAT structure and have significant downstream effects on bacterial protein expression and subsequent pathogenesis. Further study is warranted to gain a deeper understanding of the role of RNATs in bacterial physiology and the interface with the host immune response. Greater knowledge of the role of RNATs in bacterial pathogenesis may reveal novel strategies to target these structures for therapeutic treatment.

## Materials and methods

### Cell lines and primary cell cultures

HT-29 cells (a kind gift from Dr. Jan Carette, Stanford University) were maintained in DMEM medium supplemented with L-glutamine (Thermo Fisher Scientific #11995073) and 10% heat-inactivated fetal bovine serum (FBS) and split every two days or as needed. For infection experiments, HT-29 cells were seeded at $5x10^4$ cells per well in 96-well plates. 24 hours later, the

media was removed and replaced with DMEM supplemented with L-glutamine without FBS in order to serum-starve the cells for 24 hours prior to infection.

THP-1 cells were maintained in RPMI medium supplemented with L-glutamine (Thermo Fisher Scientific #11875119) and 10% heat-inactivated FBS and split every two days or as needed. To differentiate to macrophages for inflammasome activation experiments, THP-1 cells were seeded at $5x10^4$ cells per well in 96-well plates in the presence of 100 nM PMA (Invivogen #tlrl-pma). 24 hours later, the media was removed, each well was washed once with RPMI, and the media was replaced with RPMI supplemented with L-glutamine and 10% FBS. The cells were used for infections after 3 days of differentiation. Prior to infection, the differentiated THP-1 cells were primed for 18–24 hours with 100 ng/ml LPS (Invivogen #tlrl-3pelps).

Primary human monocyte-derived macrophages were derived from peripheral blood mononuclear cells (PBMCs) isolated from room temperature LRS chambers of de-identified same-day blood draws purchased through the Stanford Blood Center. Blood was flushed from the LRS chamber using room temperature 1x phosphate buffered saline (PBS) pH 7.4. The blood-PBS solution was carefully overlaid onto room temperature Ficoll Paque Plus (Fisher Scientific #45-001-749) and then centrifuged at 900 x*g* for 30 minutes at room temperature in a swing bucket centrifuge with no brakes. Following centrifugation, the buffy coat containing PBMCs was harvested, transferred to new 50 ml conicals, and the volume was brought up to 50 ml with room temperature PBS. This mixture was then centrifuged at 250 x*g* for 10 min at room temperature in a swing bucket centrifuge with half brake. Following centrifugation, the supernatant was aspirated off and the cells were resuspended in 50 ml room temperature PBS to wash, followed by centrifugation at 250 x*g* for 10 minutes with high brake. This wash step was repeated until the supernatant was clear, indicating an absence of platelets. Following washes, the PBMCs were resuspended in serum-free RPMI and viable cells were counted using 0.2% Trypan blue staining and a hemacytometer. $2x10^8$ PBMCs were plated in non-tissue culture 15 cm Petri dishes and incubated at 37˚C in 5% $CO_2$ for 3 hours to allow cells to adhere. After 3 hours, the media was aspirated, and the plates were washed 3 times with room temperature PBS. The media was then replaced with RPMI+10% FBS+30 ng/ml human M-CSF (hM-CSF; PeproTech #300-25-10). Two days after isolation, media was removed, the cells were washed twice with room temperature PBS, and fresh RPMI+10% FBS+30 ng/ml hM-CSF was added. At six days post-isolation, the differentiated macrophages were harvested and plated for infection experiments. The media was removed and the cells were washed once with room temperature PBS. Ice-cold PBS was then added to each dish of cells and the dishes were incubated on ice for 20 minutes. Cells were lifted by gently scraping. The cell suspension was then centrifuged at 250 x*g* at 4˚C for 20 minutes and the supernatant was aspirated. The cell pellets were resuspended in RPMI+10% FBS and counted using a Bio-Rad TC20 cell counter. Human monocyte-derived macrophages (hMDMs) were plated at the following densities for infection experiments in RPMI+10% FBS+30 ng/ml hM-CSF: $5x10^4$ cells per well in 96-well plates (LDH, IL-1β ELISA, and IncuCyte) and $2x10^6$ cells per well in 6-well plates (Western blot). Prior to infection, the hMDMs were primed for 18–24 hours with 100 ng/ml LPS.

## Bacterial cultures

Bacterial strains used in this study are listed in S2 Table. *E. coli* were grown in Luria Bertani (LB) medium (10 g/L tryptone+5 g/L yeast extract+10 g/L NaCl) at indicated temperatures. If required, antibiotics were added to the media at the following final concentrations: ampicillin

or carbenicillin, 100 μg/ml. For the induction of the $P_{BAD}$ promoter, the medium was supplemented with L-arabinose to a final concentration of 0.01% (w/v).

*S.* Typhi strains were routinely cultured in LB medium without antibiotic selection. Prior to each experiment, *S.* Typhi strains were struck out from glycerol stocks onto LB agar plates without antibiotic selection and grown overnight at 37˚C. For culture conditions specific to particular experiments, please see relevant section below. Since *S.* Typhi can lose the *viaB* locus genes through lab passaging, *S.* Typhi strains were routinely tested for retention of these genes by performing Vi agglutination assays and PCR. To test for Vi agglutination, *S.* Typhi strains were grown overnight in LB broth at 37˚C shaking at 200 rpm. In the morning, they were back-diluted 1:50 into low salt (5 mM NaCl) LB broth and grown for a further 3 hours at 37˚C at 200 rpm. *Salmonella* Vi antiserum (BD Difco #228271) was added to the culture at a 1:100 dilution. After a 10 minute incubation, agglutination was assessed visually under a microscope.

Colony PCRs were routinely performed on freshly struck out *S.* Typhi strains to verify retention of *viaB* genes. One bacterial colony was suspended in 100 μl 1x PBS pH 7.4 and incubated at 95˚C for 10 minutes before being used as the DNA template in PCR reactions using GoTaq polymerase (Promega #M3008) and primers oSMB64/65 and oSMB75/76 (see S3 Table). PCR products were run on a 0.9% agarose TAE gel and visualized on a Bio-Rad ChemiDoc.

For low temperature culturing conditions, *E. coli* or *S.* Typhi strains were grown at ambient temperature. The average ambient temperature ranged from 23˚ to 25˚C, as indicated in each figure.

## Secondary structure prediction

RNA secondary structures were computed via Mfold web server using the default settings [36].

## pBAD plasmid construction

Oligonucleotides and plasmids utilized in this study are summarized in S3 or S4 Tables, respectively. Enzymes for cloning were obtained from Thermo Scientific (*Nhe*I #ER0971, *Eco*RI #ER0271, *Eco*RV #ER0301, *Sma*I #ER0661). PCR, DNA manipulations, DNA restrictions, and transformations were performed according to standard protocols [60]. Point mutations were generated by site-directed mutagenesis according to the instruction manual of the QuikChange mutagenesis kit (Agilent Technologies #210505). All constructed plasmids were confirmed by restriction analysis and automated sequencing (Eurofins Genomics, Ebersberg, Germany).

The *tviA* 5'-UTR:*bgaB*-His fusion plasmid pBO4421 was constructed as follows: First, a 154 bp fragment was amplified via PCR using primer pair tviA_NheI_fw/tviA_EcoRI. Next, the resulting PCR-product was ligated into the *Nhe*I and *Eco*RI restriction sites of pBAD2-*bgaB*-His resulting in pBO4421 [42]. Mutations rep3 (T90,92C), derep4 (T89,91G,C93G), rep1 (T90C), and rep2 (T92C) were inserted into pBO4421 via site-directed mutagenesis using primer pairs tviA_rep3_fw/ tviA_rep3_rev, tviA_derep4_fw/ tviA_derep4_rev, tviA_rep1_fw/ tviA_rep1_rev, and tviA_rep2_fw/ tviA_rep2_rev obtaining plasmids pBO4424, pBO4426, pBO4427, and pBO4428, respectively.

The run-off plasmid for *in vitro* transcription of the *tviA* 5'-UTR (pBO4439) was constructed by blunt-end ligation of a PCR-amplified DNA fragment (primer tviA_RO_T7_fw/ tviA_RO_NaeI_rv), comprising the T7 RNA polymerase promoter and the *tviA* 5'-UTR including 60 bp of the *tviA* coding region, into the *Sma*I restriction site of pUC18. Insertion of the stabilizing mutation rep3 (T90,92C) or destabilizing mutation derep4 (T89,91G,C93G)

into pBO4439 was achieved by site-directed mutagenesis (primer pair tviA_rep3_fw/tviA_rep3_rv or tviA_derep4_fw/tviA_derep4_rv), resulting in pBO4447 or pBO4448.

### Reporter gene activity assays

Reporter gene assays were conducted as described before [42,44]. For β-galactosidase activity assays, *E. coli* DH5α cells carrying the *tviA* 5'-UTR:*bgaB* fusion plasmids (S4 Table) were grown overnight in LB with ampicillin at 25˚C. Before being inoculated with an overnight culture to an $OD_{600}$ = 0.2, LB media supplemented with ampicillin was pre-warmed to 25˚C. After growth under continuous shaking to an $OD_{600}$ = 0.5, transcription was induced with 0.01% w/v L-arabinose. The culture was split up and shifted to pre-warmed 100 ml flasks (temperatures indicated in the respective figure). The cultures were incubated for 30 min and 400 μl samples were subsequently taken for the β-galactosidase assay. The β-galactosidase assay was carried out as described previously [61]. Standard deviations were calculated from three technical replicates.

### *In vitro* transcription

RNAs for structure probing and primer extension inhibition experiments were synthesized *in vitro* by run-off transcription with T7 RNA polymerase (Thermo Fisher Scientific #EP0111) from *Nae*I-linearized plasmids (listed in S4 Table) as previously described [42,44].

### Enzymatic structure probing

Structure probing of the 5'-UTR and 60 nt of *tviA* was performed with *in vitro* transcribed RNA using pBO4439, pBO4447, and pBO4448 as template. The *in vitro* transcribed RNA was purified and dephosphorylated using calf intestinal phosphatase enzyme (Thermo Fisher Scientific #18009019 or New England Biolabs #M0290). The RNA was labeled with [$^{32}$P] at the 5′ end as described elsewhere [62]. Partial digestions of radiolabeled RNA with ribonuclease T1 (0.0025 U; cuts 3' of unpaired guanines; Thermo Fisher Scientific #EN0541 or Invitrogen Ambion #AM2280) and T2 (0.053 U; cuts 3′ of unpaired nucleotides with a slight preference for adenines; MoBiTec #GE-NUC00400-02) were performed according to Waldminghaus et al. (2007) at 25, 37, and 42˚C. For digestion with RNase T1 and RNase T2 a 5 x TN buffer (100 mM Tris acetate, pH 7, 500 mM NaCl) was used. An alkaline hydrolysis ladder was prepared as described before [62]. The T1-ladder was generated by using 30000 cpm labeled RNA. The RNA was heated with 1 μl sequencing buffer (provided with RNase T1) at 90˚C. Subsequently, the RNA was incubated with the enzyme at 37˚C for 5 min. All resulting fragments were separated by size on an 8% polyacrylamide gel.

### Toeprinting

Primer extension and primer extension inhibition analysis (i.e., toeprinting) analysis was performed with 30S ribosomal subunit, *in vitro* transcribed RNA and tRNA$^{fMet}$ (Sigma Aldrich #63231-63-0) according to a protocol described before [46]. A 5'-[$^{32}$P]-labeled *tviA*-specific oligonucleotide tviA_RO_NaeI_rv, complementary to nucleotides +49 to +60 (from ATG) of the *tviA* mRNA, was used as a primer for cDNA synthesis. The radiolabeled primer (0.16 pmol) was annealed to the *tviA* mRNA (0.08 pmol) and incubated with 30S ribosomal subunit (24 pmol) or Watanabe buffer (60 mM HEPES/KOH; 10.5 mM Mg(COO)$_2$; 690 mM NH$_4$COO; 12 mM β-mercaptoethanol; 10 mM spermidine; 0.25 mM spermine) in the presence of tRNA$^{fMet}$ (8 pmol) at 25, 37 or 42˚C for 10 min. The 30S ribosomal subunit was purified from *E. coli* cells via density gradient centrifugation or FPLC. After addition of 2 μl

MMLV-Mix (VD+Mg$^{2+}$ buffer, BSA, dNTPs and 800 U MMLV reverse transcriptase; Affymetrix #78306), cDNA synthesis was performed for 10 min at 37˚C. The reaction was stopped by addition of formamide loading dye and samples were separated on an 8% denaturing polyacrylamide gel. The Thermo Sequenase Cycle Sequencing Kit (Affymetrix #78500) was used for sequencing reactions with plasmid pBO4439 as template and radiolabeled primer tviA_RO_NaeI_rv.

### Band intensity quantification

To quantify band intensities obtained by enzymatic structure probing as well as toe printing experiments we used the SAFA package [63]. These data (peak areas) obtained from structure probing gel quantification were processed as follows. Prior to normalizing each RNase T1 or T2 signals to the respective maximum signal, band signals resulting from RNA degradation were excluded by subtracting each peak area (band intensity) measured in the control lane (minus RNase) from signals of the respective band in the RNase T1/T2 probing lanes. Toe printing signals were corrected by background signals (signals in -30S lanes) and each signal was normalized to the peak areas detected at 25˚C.

### S. Typhi strain construction

Both gene knockouts and point mutants in *S*. Typhi were generated using the scarless genome editing technique described [64]. To transform in the helper plasmid pSLTS, which encodes the meganuclease I-SceI under the control of the anhydrotetracycline-inducible *tetA* promoter and the lambda red recombinase genes under the control of the arabinose-inducible *araB* promoter and carries a temperature-sensitive origin of replication, an overnight culture of *S*. Typhi Ty2 was back-diluted into no salt LB broth and grown to an OD$_{600}$ of ~0.4–0.8 before washing twice with ice-cold ddH$_2$O. Washed cells were resuspended in ice-cold ddH$_2$O and 100 ng of pSLTS plasmid was electroporated into 50 µl of resuspended bacteria. *S*. Typhi (pSLTS) was recovered in SOC medium shaking at 30˚C before plating on LB+50 µg/ml carbenicillin plates and overnight growth at 30˚C.

Mutation cassette plasmids were generated using Gibson Assembly (New England Biolabs #E2611S). The mutation cassettes consisted of Ty2 DNA sequence flanking an antibiotic selection cassette for homologous recombination into the *S*. Typhi Ty2 chromosome. The Ty2 genomic DNA regions were either obtained as gBlocks (IDT DNA; used to make pSMB11) or amplified from Ty2 genomic DNA using PCR (Phusion High-Fidelity DNA polymerase, Thermo Fisher Scientific #F530L) using primers that contained hanging overlap regions for Gibson assembly. The selection cassette containing the antibiotic marker Cm$^R$ and I-SceI cut site was amplified from pT2SC via PCR (Phusion High-Fidelity DNA polymerase). The plasmid backbone fragment was amplified via PCR (Phusion High-Fidelity DNA polymerase) from pUC19. Following amplification, all PCR products were gel-purified (Qiagen QIAquick Gel Extraction Kit, Qiagen #28706) prior to Gibson assembly. See S3 Table for primers used and S4 Table for mutation cassette plasmid details. Gibson assembly products were transformed into NEB 5-alpha competent *E. coli* (New England Biolabs #C2987H) and constructs were screened for correct assembly using Sanger sequencing (ElimBio, Hayward, CA, USA) and primers pHA.seq.F and pHA.seq.R. To generate plasmid pSMB12, plasmid pSMB11 was used as the template to perform site-directed mutagenesis (QuikChange Lightning Multi Site-Directed Mutagenesis Kit; Agilent #210505) with primers oSMB113 and oSMB114 following the manufacturer's instructions.

Following sequencing verification, the mutation cassette (i.e., the plasmid region containing Ty2 genomic DNA sequences flanking the Cm$^R$ and I-SceI cut site) was PCR-amplified using

Phusion polymerase and gel-purified (See S3 Table for primers used to amplify mutation cassettes and S4 Table for descriptions of mutation cassette plasmids). An overnight culture of *S.* Typhi(pSLTS) in no salt LB broth+50 μg/ml carbenicillin grown at 30˚C at 200 rpm was back-diluted 1:100 into no salt LB broth+50 μg/ml carbenicillin and grown shaking at 30˚C at 200 rpm. One hour after back-dilution, L-arabinose was added to a final concentration of 2 mM to induce expression of the lambda red recombinase, and the culture was incubated for a further 2–3 hours until the $OD_{600}$ reached ~0.7–0.9. The culture was washed twice with ice-cold $ddH_2O$ before being resuspended in ice-cold $ddH_2O$ and then electroporated with 1 μg of the mutation cassette. Transformed bacteria were recovered in SOC shaking at 30˚C before plating on LB+50 μg/ml carbenicillin+20 μg/ml chloramphenicol plates and overnight growth at 30˚C. Transformants were restruck onto LB+50 μg/ml carbenicillin+20 μg/ml chloramphenicol plates and grown overnight at 30˚C. Restruck colonies were suspended in 250 μl 1x PBS pH 7.4, spread onto LB+50 μg/ml carbenicillin+100 ng/ml anhydrotetracycline plates, and grown overnight at 30˚C to induce I-SceI expression to loop out the selection cassette. Colonies were then repatched twice onto LB plates without any antibiotic selection and grown at 37˚C to cure of pSLTS. To verify loss of pSLTS and the selection cassette, isolates were tested for growth on LB+50 μg/ml carbenicillin and LB+20 μg/ml chloramphenicol, respectively, at both 30˚C and 37˚C. Sensitive isolates were then screened for the desired mutation by PCR and/or Sanger sequencing (see S3 Table for PCR and sequencing verification primers). All verified strains were stored in 15% glycerol at -80˚C.

## Microscopy and flow cytometry

*S.* Typhi strains were grown in LB broth for 6 hours shaking at 200 rpm at 37˚C and then transferred to an ambient temperature of 23˚C for overnight growth shaking at 200 rpm. In the morning, the cultures were back-diluted into low salt (5 mM NaCl) LB broth to an $OD_{600}$ of 0.2 and grown shaking at 200 rpm at 23˚C until the $OD_{600}$ reached 0.5–0.6. Cultures were then shifted to 37˚C or 42˚C or left at 23˚C while shaking for 30 minutes to 2 hours, as indicated in the figures.

Following the temperature shift, cultures were transferred to ice and then harvested by spinning at 6000 x*g* at 4˚C for 5 min and washed once in 1x PBS pH 7.4. Bacteria were fixed in 4% paraformaldehyde in 1x PBS pH 7.4 for 30 minutes at room temperature. Following two washes with 1x PBS pH 7.4, bacteria were incubated with *Salmonella* Vi antiserum (BD Difco #228271) diluted 1:250 in 1x PBS pH 7.4 for one hour at room temperature. The bacteria were washed twice with PBS and then stained with anti-rabbit-488 (Thermo Fisher Scientific #A-11008) diluted 1:500 in 1x PBS pH 7.4 for one hour at room temperature in the dark. Following secondary antibody staining, bacteria were washed two more times. For flow cytometry analysis, bacteria were resuspended in 1x PBS pH 7.4+2mM EDTA, and run on an Accuri C6 Plus (BD) flow cytometer. Data were analyzed by FlowJo (FlowJo LLC). For immunofluorescence microscopy analysis, bacteria were added to poly-L-lysine-coated glass coverslips and incubated for 30 minutes in the dark. The coverslips were gently washed twice with PBS before being mounted on glass microscope slides with ProLong Diamond Antifade Mountant (Thermo Fisher Scientific #P36970). The slides were cured for 24 hours in the dark at room temperature before being sealed with clear nail polish and stored at -20˚C until imaging on a Zeiss LSM 700 confocal microscope with the ZEN 2010 software. Images were processed using Fiji software (i.e., ImageJ).

## Motility plate assay

For each assay, fresh low agar concentration plates (0.3% bacto-agar and 10 g/L bacto-tryptone) were made with either 0 mM NaCl or 150 mM NaCl the day before inoculation.

Immediately prior to inoculation, the plates were dried in a biosafety cabinet for 20 minutes to reduce excess surface water. Overnight cultures of *S*. Typhi grown in LB broth were pelleted at 4500 x*g* for 5 minutes and then resuspended in 1x PBS pH 7.4. The cultures were diluted to an $OD_{600}$ = 3.0, and the center of each motility plate was inoculated with 5 μl of the respective diluted strain. The inoculum was allowed to fully dry at room temperature before the plates were placed in a 37˚C incubator for 24 hours. Halo diameters were measured with a ruler and images of the plates were captured on a Bio-Rad ChemiDoc.

## Bacterial RNA extraction and qRT-PCR

*E. coli* DH5$\alpha$ containing the pBAD-*bgaB* plasmid with different versions of the *tviA* 5' UTR were grown as described for the β-galactosidase activity assay. *S*. Typhi strains were grown as described for the Vi capsule expression analysis via flow cytometry. Following temperature shifts, 1.5 ml of each culture was pelleted at 6000 x*g* at 4˚C for 5 minutes. The supernatant was discarded and each pellet was resuspended in 500 μl of TRIzol reagent (Thermo Fisher Scientific #15-596-018) and stored at -80˚C until RNA extractions could be performed.

To extract RNA, samples were thawed on ice and 100 μl of chloroform was added to each sample before mixing by inversion and centrifugation at 20,000 x*g* at 4˚C for 15 minutes. The aqueous phase was transferred to a new tube and then 150 μl of 70% ethanol was add to each sample. The RNA-ethanol mixture was applied to a column from the RNeasy Mini Kit (Qiagen #74106) and centrifuged at 9400 x*g* at room temperature for 1 min. The RNeasy Mini Kit was then used to complete the extraction with the following modifications. The column was washed with 350 μl of RW1 buffer and spun at 9400 x*g* at room temperature for 1 min. 80 μl of RNase-free DNase in RDD buffer was add to each column and the columns were incubated for 15 minutes at room temperature. Another 350 μl RW1 was added to the column and the spin was repeated. This was followed by two washes with 500 μl buffer RPE. A final spin was performed to remove residual RPE buffer before the column was transferred to a clean tube and the RNA was eluted in 35 μl Ultrapure RNase- and DNase-free water. RNA concentration was determined using a Nanodrop spectrophotometer and the samples were stored at -80˚C. cDNA was generated using the SuperScript III First-Strand Synthesis System (Thermo Fisher Scientific #18080051). The same amount of RNA was added to each reaction to normalize across samples and the manufacturer's protocol was followed using random hexamers. Synthesized cDNA was stored at -20˚C.

To assess expression of bacterial genes, FastStart Universal SYBR Green Master mix (Sigma Aldrich #4913850001) was used in combination with the synthesized cDNA and primers listed in S3 Table. The reactions were run on a Bio-Rad CFX Connect. Relative expression values were calculated using the ΔΔCt method. Each experimental value was first normalized to the housekeeping control (i.e., 16S rRNA) for that particular sample, followed by normalization to the average of a control condition (i.e., the low temperature WT condition).

## HT-29 infections

*S*. Typhi strains were cultured overnight in LB broth at 37˚C shaking at 200 rpm. In the morning, the cultures were back-diluted 1:50 into low salt (5 mM NaCl) LB broth and grown at 37˚C shaking at 200 rpm for 2–3 hours until the $OD_{600}$ reached 0.6–0.8 for non-invasive conditions. The bacteria were centrifuged at 4500 x*g* at room temperature for 5 minutes. The supernatant was removed and the bacteria were washed once with 1x PBS pH 7.4 before a final resuspension in PBS and $OD_{600}$ measurement. The bacteria were diluted in DMEM to infect at an MOI of 25. Media was removed from the serum-starved HT-29 cells, the cells were washed once with DMEM, and then the cells were infected with the diluted bacteria in DMEM. Immediately after adding the bacteria to the cells, the plate was spun at 250 x*g* for 5

minutes at room temperature before placement in a 37˚C 5% $CO_2$ incubator for 1 hour. After one hour, the media was removed and the cells were washed twice with PBS. The media was then replaced with DMEM supplemented with L-glutamine, 2% heat-inactivated FBS, and 100 ug/ml gentamicin. After one hour, the media was removed and the cells were washed twice with PBS before the media was replaced with DMEM supplemented with L-glutamine, 2% heat-inactivated FBS, and 10 ug/ml gentamicin. At 6 and 18 hours post-infection, the supernatants from each well were transferred to a new 96-well plate and stored at -20˚C until IL-8 secretion could be assessed following the manufacturer's instructions for a human IL-8 ELISA (Thermo Fisher Scientific #88-8086-22).

## THP-1 and hMDM infections

*S.* Typhi strains were cultured overnight in LB broth at 37˚C shaking at 200 rpm. In the morning, the cultures were back-diluted 1:50 into high salt (300 mM NaCl) LB broth and grown at 37˚C shaking at 200 rpm for 2–3 hours until the $OD_{600}$ reached 0.6–0.8 for invasive (i.e., *Salmonella* Pathogenicity Island-1-inducing) conditions. The bacteria were centrifuged at 4500 x*g* at room temperature for 5 minutes. The supernatant was removed and the bacteria were washed once with 1x PBS pH 7.4 before a final resuspension in PBS and $OD_{600}$ measurement. The bacteria were diluted in RPMI without phenol red (Thermo Fisher Scientific #11835030) to infect at an MOI of 10 or 25, as indicated in figure legends. Media was removed from the differentiated and LPS-primed THP-1 or hMDM cells, the cells were washed once with RPMI without phenol red, and the cells were infected with the diluted bacteria in phenol red-free RPMI supplemented with L-glutamine and 10% FBS. For kinetic cell death experiments, SYTOX Green nucleic acid stain (Thermo Fisher Scientific #S7020) was also added to the media at a final concentration of 20 nM. For Western blot sample infections, RPMI media without FBS was used for the infection. Immediately after adding the bacteria to the cells, the plates were spun at 250 x*g* for 5 minutes at room temperature before placement in a 37˚C 5% $CO_2$ incubator for 1 hour. For kinetic cell death experiments, the plate was placed in an IncuCyte S3 (Essen Bioscience) in a 37˚C 5% $CO_2$ incubator and two phase and green fluorescence images were taken per well with the 10x objective every 30 minutes. After one hour, gentamicin was added to a final concentration of 10 μg/ml and the plates were returned to the incubator.

At 2 hours post-infection, 50 μl of supernatant from each well was transferred to a new 96-well plate and stored at -20˚C until IL-1β levels could be assessed following the manufacturer's instructions for a human IL-1β ELISA (Thermo Fisher Scientific #88-7261-88). Another 50 μl of supernatant was used to assess LDH release as a measure of pyroptotic cell death following the manufacturer's instructions for the CytoTox 96 Non-Radioactive Cytotoxicity Assay (Promega #G1780). Absorbance for the LDH assay was measured at 490 nm on a BioTek Synergy HTX. A "% cell death" was calculated as follows:

$$\frac{Experimental\ absorbance - Untreated\ absorbance\ average}{Lysis\ absorbance\ average - Untreated\ absorbance\ average} * 100$$

For the kinetic cell death experiments, the IncuCyte S3 software was used to analyze SYTOX Green staining. At the beginning of the infection, a set of uninfected wells were treated with 0.1% Triton X-100 to kill all the THP-1 cells and obtain a "100% cell death" SYTOX Green count. A "% cell death" was calculated as follows:

$$\frac{Experimental\ condition\ SYTOX\ Green\ count}{100\%\ cell\ death\ SYTOX\ Green\ count} * 100$$

## Western blots

For hMDM Western blot samples, at 2 hours post-infection the supernatants were removed and 20% TCA was added to the supernatants to achieve a final concentration of 10%. The supernatant proteins were precipitated on ice for one hour before centrifugation for 30 minutes at 4˚C at 20,000 x*g*. Following centrifugation, the supernatant was removed and the pellets were washed with ice-cold acetone. The spin was repeated for 10 minutes, after which the supernatant was removed and the pellets were allowed to air dry for 10–20 minutes at room temperature. To harvest the cell lysate proteins, 100 μl of RIPA buffer (50 mM Tris HCl pH 7.4, 150 mM NaCl, 0.1% SDS, 0.5% sodium deoxycholate, 1% Triton X-100) with cOmplete Mini Protease Inhibitor Cocktail (Sigma Aldrich #11836153001) was added to each well after the supernatants were removed for protein precipitation. The plates were incubated on ice for 10 minutes followed by scraping and transfer of the cell lysate to 1.5 ml Eppendorf tubes. The samples were incubated on ice for 30 minutes and then centrifuged for 15 minutes at 4˚C at 20,000 x*g*. Following centrifugation, the supernatant was used to resuspend the TCA-precipitated proteins to obtain a cell lysate+supernatant protein sample, and the samples were stored at -20˚C. Prior to running the samples on a gel, total protein was determined for each sample using the Pierce Coomassie (Bradford) Protein Assay Kit (Thermo Fisher Scientific #23200). The samples were all normalized to the same protein concentration, 6x Laemmli buffer (final concentration 1x) with β-mercaptoethanol (final concentration 2%) was added, and the samples were boiled for 10 minutes at 95˚C.

For *S*. Typhi Western blot samples, strains were cultured statically for 24 hours under the following conditions: low salt media (5 mM NaCl LB broth) at 23˚C, low salt media at 37˚C, high salt media (300 mM NaCl LB broth) at 23˚C, or high salt media at 37˚C. To harvest, the $OD_{600}$ of each culture was measured and $1 \times 10^9$ CFU of each culture was transferred to a fresh tube and spun at 3200 x*g* for 20 minutes at 4˚C. The supernatant was discarded, and each bacterial pellet was resuspended in 100 μl of 1x Laemmli buffer diluted in PBS and supplemented with 5% β-mercaptoethanol. The samples were stored at -20˚C and boiled for 10 minutes at 95˚C prior to running on a gel.

The hMDM samples were run on 4–20% Mini-PROTEAN TGX Precast protein gels (Bio-Rad #4561093). The *S*. Typhi samples were run on Criterion 26-well (Bio-Rad #3459903) gels made with the 10% TGX FastCast acrylamide gel kit (Bio-Rad #1610173). The Precision Plus Protein Dual Color Standard (Bio-Rad #161–0394) was used for all gels, and they were blotted onto Trans-blot Turbo PVDF membranes (Bio-Rad #1703127) using a Trans-blot Turbo machine (Bio-Rad). For the *S*. Typhi Western blots, prior to antibody probing total protein was assessed using SYPRO Ruby Protein Blot Stain (Bio-Rad #1703127) according to the manufacturer's protocol. Membranes were blocked with PBS+0.1% Tween 20 (PBST)+5% milk for 1 hour shaking at room temperature followed by overnight incubation with primary antibody shaking at 4˚C. Primary antibody was washed off with PBST shaking at room temperature for 5 minutes for a total of three washes. Membranes were incubated with secondary antibody for 1 hour shaking at room temperature followed by three washes with PBST. SuperSignal West Femto Maximum Sensitivity Substrate (Thermo Fisher Scientific #34096) was then added to each blot and chemiluminescent signal was visualized on a Bio-Rad ChemiDoc. Prior to probing the membrane with the next primary antibody, it was stripped using Re-blot Plus Mild Antibody Stripping Solution (Sigma Aldrich #2502) while shaking at room temperature for 15 minutes. Primary antibodies used for hMDM samples: anti-human caspase-1 (R&D Systems #AF6215), anti-human IL-1β (Abcam #ab2105), anti-human cleaved N-terminal GSDMD (Abcam #ab215203), and anti-β-actin (Sigma Aldrich #A1978). Primary antibodies used for *S*. Typhi samples: adsorbed *Salmonella* H antiserum d (BD Difco #228231) and anti-*E. coli* RecA

(Bio Academia #61–003). Secondary antibodies used: anti-goat IgG, HRP-linked (Thermo Fisher Scientific #61–1620), anti-rabbit IgG, HRP-linked (GE Healthcare #NA934V), and anti-mouse IgG, HRP-linked (GE Healthcare #NA931V).

## Antibody adsorption

To reduce nonspecific antibody staining and increase specificity for flagellin, we cleaned up the *Salmonella* H antiserum d (BD Difco #228231) via adsorption. A 9 ml overnight culture of *S.* Typhi strain SMB24 (Δ*fliC*) in LB broth had formalin added to a final concentration of 0.5%. This mixture was incubated at room temperature with shaking for 24 hours to fix the bacteria. Once fixed, the bacteria were split between 5 tubes and spun at 8000 x*g* at 4˚C for 5 minutes, followed by two washes with 1x PBS pH 7.4. After discarding the supernatant from the second wash, one of the fixed bacterial pellets was resuspended in 200 μl of the *Salmonella* H antiserum d and incubated on ice for 30 minutes. Following incubation, the tube was centrifuged at 10,000 x*g* at 4˚C for 5 minutes and the supernatant was used to resuspend another pellet of fixed bacteria. This incubation and supernatant transfer process was repeated with all 5 tubes of fixed bacterial pellets. Following the final incubation, the supernatant was transferred to a new tube and stored at -20˚C until use for Western blotting.

## Sequence alignment

For the alignment of *tviA*/*vipR* 5' UTR sequences, nucleotide sequence information for each *Salmonella* strain and *Citrobacter freundii* was downloaded from NCBI or BioCyc (https://biocyc.org/). Benchling was used to create and export an alignment of the sequences.

## Statistics

Prism v8.4.3 (GraphPad Software, LLC) was used to create graphical figures and perform statistical analyses. Statistical tests performed and corresponding details are indicated in the figure legends.

The numerical data used in all figures are included in S1 Data.

## Supporting information

**S1 Data. Excel spreadsheet containing numerical data, in separate sheets, for Figure panels 1c, 1d, 2c, 2d, 2e, S4b, S4c, 4b, 4c, 5a, 5b, 5d, 5f, S5b, 6a, 6b, 7a, 7b, 7c, 7d, 7e, 7f, 7g, 7h, S6.** (XLSX)

**S1 Table. Related to Fig 3.** Hard constraints for RNA structure prediction. (DOCX)

**S2 Table. Bacterial strains used in this study.** (DOCX)

**S3 Table. Oligonucleotides used in this study.** (DOCX)

**S4 Table. Plasmids used in this study.** (DOCX)

**S1 Fig. Mechanisms of RNAT- and TviA-mediated regulation.** A) Schematic depicting RNAT function in response to temperature. At low temperatures, Watson-Crick base-pairing in the 5' UTR of the mRNA generates secondary structure that prevents ribosome access to the SD region and translation. As the temperature increases, the nucleotide bonds melt until the

structure is fully open, ribosomes can bind, and translation can occur. B) Schematic depicting effect of osmolarity on TviA production and subsequent virulence factor expression. Under high osmolarity conditions (e.g., intestinal lumen), transcription of *tviA* is repressed, meaning that *S*. Typhi has high expression of flagellin and type 3 secretion system-1 (T3SS-1) but does not express Vi capsule. This expression pattern renders *S*. Typhi motile and invasive. Transition to low osmolarity conditions (e.g., intestinal epithelium or intracellular) induces *tviA* transcription. TviA then induces Vi capsule expression and suppression of flagellin and T3SS-1 expression, which gives *S*. Typhi an immune evasive phenotype.
(TIF)

**S2 Fig. Related to Fig 1.** Example of gating strategy used to identify *S*. Typhi bacteria and quantitate level of Vi capsule expression following temperature shifts.
(TIF)

**S3 Fig. Related to Fig 2.** A and B) Schematic representation of the secondary structure of the *tviA* RNAT stem (nucleotides 85 to 132 plus AUG codon) including the fourU motif (4U; blue nucleotides), the SD region (black nucleotides), and the AUG codon (orange nucleotides). The desired (A) as well as the Mfold-predicted minimum free energy (B) repressed (mutations highlighted in red) and derepressed (mutations highlighted in green) RNAT structures resulting from nucleotide exchanges are depicted as well. C) Effect of altering base-pairing of the fourU region on temperature-dependent translation. Comparison of translation efficiency at different temperatures using the *S*. Typhi wildtype *tviA* 5' UTR (WT), the rep1 (T90C) *tviA* 5' UTR mutant, and the rep2 (T92C) *tviA* 5' UTR mutant *bgaB* fusion constructs. Data shown are representative of 4 independent experiments (C) with triplicate samples for each condition. Data are represented as mean ± SD. NS, not significant. Statistical significance determined using two-way ANOVA with Tukey's correction. $^{*}$ p < 0.05, $^{****}$ p < 0.0001.
(TIF)

**S4 Fig. Related to Fig 3.** A) Entire gel from enzymatic structure probing shown in Fig 3A. 5' end-labeled *in vitro* transcribed RNA containing the wildtype *tviA* 5' UTR (WT), the rep3 (T90,92C) *tviA* 5' UTR mutant (REP), or the derep4 (T89,91G;C93G) *tviA* 5' UTR mutant (DEREP) was enzymatically probed with RNases T1 (cuts 3' of single-stranded guanines) and T2 (cuts 3' of single-stranded nucleotides with preference order: A > C > U > G) at 25, 37, and 42˚C. Fragmented RNA was separated on an 8% polyacrylamide gel. AL: alkaline ladder. $L_{T1}$: RNase T1 cleavage in sequence buffer at 37˚C. C: RNA treated with water instead of RNase at 42˚C. For many residues, the cleavage pattern fit the predicted secondary structure, with nucleotides susceptible to RNase T1 and T2 cleavage at 25˚C (see also Fig 3B). These nucleotides are frequently located within predicted hairpin loops (e.g., GGG26-28, UAGG106-109; green highlight) or associated with inter-loop regions (e.g., GA74-75, GG97-98; orange highlight). The conformation of hairpin IB (see Fig 3B) deviates from the predicted structure because paired nucleotide stretches UG52-53 and GAAU68-71 were readily digested by the RNases at 25˚C (gray highlight). Conversely, nucleotides UG41-42, AA49-50, and GG60-61, which were predicted to be unpaired, more likely adopt a double-stranded conformation because of increased RNase-mediated cleavage at 37˚C and 42˚C (purple highlight). B) Quantification of band intensities of RNase T1/T2 cleavage products in the region of the Shine-Dalgarno sequence of the wildtype *tviA* 5' UTR at 25, 37, and 42˚C. Data shown are representative of 4 independent experiments.
(TIF)

**S5 Fig. Related to Fig 5.** A) Entire Western blot with loading controls shown in Fig 5E. Whole cell lysates of WT, Δ*fliC*, Δ*tviA*, *tviA*-REP, or *tviA*-DEREP *S*. Typhi were probed for total

protein, flagellin expression, and RecA expression after 24 hours of static growth under the following conditions: low salt media (5 mM NaCl LB broth) at 23˚C, low salt media at 37˚C, high salt media (300 mM NaCl LB broth) at 23˚C, or high salt media at 37˚C. B) Quantitation of flagellin expression from Western blot in S5A Fig. Flagellin band intensity was normalized to the RecA loading control band intensity for each sample to demonstrate similar results as seen with normalization to total protein (see Fig 5F). Data shown are representative of 2 independent experiments.
(TIF)

**S6 Fig. Related to Fig 6.** Quantitation of cleaved caspase-1 (A), cleaved IL-1β (B), and NT-GSDMD (C) from the Western blot shown in Fig 7I. Band intensities of each cleavage product were normalized to the band intensity of β-actin for each sample before being plotted. Data shown are representative of 2 independent experiments.
(TIF)

**S7 Fig. Conservation of the fourU RNAT in Vi capsule-producing, enteric fever-causing human pathogens.** Alignment of the *tviA* (or equivalent homolog *vipR* in *Citrobacter freundii*) 5' UTR nucleotide sequences from lab strain *S.* Typhi Ty2, vaccine strain *S.* Typhi Ty21a, multi-drug resistant clinical isolate *S.* Typhi CT18, clinical isolate *S.* Paratyphi C RKS4594, *S.* Dublin isolate SARB13, or opportunistic pathogen *C. freundii* reveals conservation of the fourU RNA thermosensor in enteric fever-causing human pathogens. FourU RNA thermosensor (4U) highlighted in red. Shine-Dalgarno (SD) highlighted in blue. Start codon highlighted in green. Asterisks indicate conserved residues between strains.
(TIF)

## Acknowledgments

The authors would like to thank Dr. Manuel Amieva and members of the Monack and Amieva laboratories for valuable discussions.

## Author Contributions

**Conceptualization:** Susan M. Brewer, Christian Twittenhoff, Jens Kortmann, Franz Narberhaus, Denise M. Monack.

**Data curation:** Susan M. Brewer, Christian Twittenhoff, Jens Kortmann.

**Formal analysis:** Susan M. Brewer, Christian Twittenhoff, Jens Kortmann, Franz Narberhaus, Denise M. Monack.

**Funding acquisition:** Franz Narberhaus, Denise M. Monack.

**Investigation:** Susan M. Brewer, Christian Twittenhoff, Jens Kortmann, Sky W. Brubaker, Jared Honeycutt, Liliana Moura Massis, Trung H. M. Pham.

**Supervision:** Franz Narberhaus, Denise M. Monack.

**Writing – original draft:** Susan M. Brewer, Christian Twittenhoff, Franz Narberhaus, Denise M. Monack.

**Writing – review & editing:** Susan M. Brewer, Christian Twittenhoff, Jens Kortmann, Sky W. Brubaker, Jared Honeycutt, Liliana Moura Massis, Trung H. M. Pham, Franz Narberhaus, Denise M. Monack.

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
