## [Decision Letter · Decision Letter 0]

18 Nov 2020

Dear Denise,

Thank you very much for submitting your manuscript "A Salmonella Typhi RNA thermosensor regulates virulence factors and innate immune evasion in response to host temperature" for consideration at PLOS Pathogens. As with all papers reviewed by the journal, your manuscript was reviewed by members of the editorial board and by several independent reviewers. The reviewers appreciated the attention to an important topic. Based on the reviews, we are likely to accept this manuscript for publication, providing that you modify the manuscript according to the review recommendations.

Sincerely,

Leigh Knodler

Guest Editor

PLOS Pathogens

Renée Tsolis

Section Editor

PLOS Pathogens

Kasturi Haldar

Editor-in-Chief

PLOS Pathogens

orcid.org/0000-0001-5065-158X

Michael Malim

Editor-in-Chief

PLOS Pathogens

orcid.org/0000-0002-7699-2064

Reviewer Comments (if any, and for reference):

Reviewer's Responses to Questions

**Part I - Summary**

Reviewer #1: In their study, Brewer et al. identify an RNA thermometer (RNAT) in the 5’UTR of the tviA mRNA of Salmonella Typhi, encoding a transcriptional master regulator of a well-defined regulon that comprises motility, virulence and immune evasion factors. Using a combination of genetics, RNA biochemical and phenotypic assays, the authors present mechanistic data that demonstrates that the tviA RNAT regulates translation initiation of TviA in a temperature-dependent manner, which in turn affects motility and Vi capsule expression of S. Typhi, and consequently, the host immune response to infection. Integrated with the previously known transcriptional induction of tviA through the OmpR/EnvZ two-component system, these new data suggest a compelling working model in which TviA expression is coupled to both, elevated temperature and low osmolarity, to ensure that the transition from an invasive, motile lifestyle to immune evasion occurs at the appropriate stage of the S. Typhi infection cycle.

The presented experiments were well-conducted and support the drawn conclusions. Further, the study is both, timely and relevant. I only have a few minor comments that I hope will be useful for the authors to further improve this interesting study.

Reviewer #2: Brewer et al describe the discovery and characterisation of an RNA thermosensor in the 5’ UTR region of the tviA gene of Salmonella Typhi. From the initial observation that the Vi polysaccharide expression was temperature sensitive but not as a result of transcriptional changes to the master regulator tviA, they discover a sequence with structural characteristics consistent with that of a RNAT. Through site directed mutagenesis they construct recombinant strains with altered structural properties that convincingly demonstrate that this is indeed an RNAT. The role of the RNAT in regulating expression of the viaB locus that affected the innate immune response and motility is then addressed.

This is a well written manuscript describing an excellent study that identifies a previously unknown regulatory level in the expression of an important virulence factor in S. Typhi. It is therefore highly novel and important piece of work. The only weakness is the use of three replicates throughout results in low statistical power. In general, this does not affect the major conclusions due to very low variance and large differences in the mean in most data. However, where means are similar, this may have resulted in falsely accepting the null hypothesis.

Reviewer #3: (No Response)

**Part II – Major Issues: Key Experiments Required for Acceptance**

Reviewer #1: (No Response)

Reviewer #2: (No Response)

Reviewer #3: Manuscript by Brewer et al investigates the role of 5’ UTR region of tviA, a transcriptional regulator of the critical virulence factors Vi  capsule, flagellin, and type III secretion system-1 expression, of S. Typhi. Authors identify a RNA thermosensor (RNAT) in the UTR region and show that this region post-transcriptionally controls the TviA protein. Using in vitro reporter system in E. coli and point mutations in the tviA 5’ UTR that either stabilize or destabilize the RNAT structure, the authors RNAT in UTR region is closed therefore the tviA is repressed at low temperatures. Authors also show that the RNAT melts open at higher temperatures allowing ribosome access and the translation of tviA. Authors go on to show that this system controls the expression of Vi and repression of flagella in a temperature dependent way. However, the growth in high salt (150mM NaCl) conditions can override and induce maximal suppression of flagellin expression.By utilizing mutants where the RNAT structure is stabilized or destabilized, authors show that the regulation of flagellin expression would be detected by the host inflammasome in THP-1 macrophages. Overall this is an elegant investigation of the temperature dependent regulation of tviA. The manuscript is well written. My only concern is the big conclusion drawn in Fig 8 is not fully supported. Although the RNAT function was investigated in high and low temperature or in high and low osmolarity, these experiments were done separately. To conclude the combinatorial effect of these conditions, authors should perform the experiments in the four conditions as proposed in Fig 8, high osmolarity and low temperature, high osmolarity and high temperature, low osmolarity and low temperature, low osmolarity and high temperature. There are no in vivo experiments to support the hypothesis in Fig 8 but at least if the in vitro experiments can mimic what might be happening in vivo, this would increase the impact of the findings

**Part III – Minor Issues: Editorial and Data Presentation Modifications**

Reviewer #1: • Experimental data were typically collected over several independent experiments, each performed in technical replicates. However, it seems as if for the presentation of the data often times just one (representative) biological replicate with all its technical replicates was shown (e.g. Fig. 1). Unless there is a reason against it (e.g. large donor variability for the assays with primary human macrophages), I would however prefer plotting individual biological – rather than technical – replicates.

• Fig. 2E: Do the authors have an explanation for the (RNAT-independent) increase of bgaB mRNA levels at 42°C? Unless the pBAD promoter responds to temperature, this would imply that the mRNA is more stable in the warmth.

• Fig. 7A, D, E: Some data points are below the 0% line. How can the percentage of cell death be negative?

• Western blot in Fig. 7I: The effect of mutating the tviA RNAT (‘tvi-REP’) or deleting the entire gene (‘ΔtviA’) is less clear on the CASP1 and IL1b band signals than it is on GSDMD. Could the authors quantify band intensities for this western blot (as they do for the toeprinting assay in Fig. 4B, C)?

Text edits:

• Line 159: In the main text, the authors use a different numbering system (“position 123 to 127 within the 5’UTR”) than in the referenced figure (“-132 nt to +3 nt from AUG”; line 170 and Fig. 2A). For reasons of consistency, this may be homogenized.

• Lines 184 and 375: Please correct “16S mRNA” to “16S rRNA”.

• Lines 279, 281, 295, 296: The authors may want to label stems/hairpins I and II in Fig. 2A to allow the readers to better follow these text parts.

• Lines 339-341: Please specify to which UTR variant the quantifications in B and C refer. (I assume it’s the wild-type, but since there are additional UTR variants depicted on the gel in panel A, it should probably be mentioned explicitly.)

• The names of the bacterial mutant strains in Fig. 7D and E should be italicized (as done in the remaining panels of this figure).

Reviewer #2: General comment

1. Data in figure 1 includes treatment at 23C and later experiments equivalent low temperature treatments are at 25C. Is this a typo, or were the experiments really performed at different temperatures and if so was there a rationale for this change?

2. At what temperature does the RNAT de-repress translation of TviA?

Specific comments

3. 156 'S' should be italics.

4. 156-158. Please describe the use of Mfold in Materials and Methods.

5. 162-167. Please make it clear why these structural elements were indicative of a cis-regulatory thermosensor. Suggest citing studies in which similar structures were shown to have these functions.

6. 588-590. These are new data analysis and I suggest that they are included in the results section. The tviA 5' UTR of S. Paratyphi C and S. Dublin are included in the alignment in Figure S5 and should also be mentioned in the text. Currently only the presence of the genes in these serovars is mentioned.

7. 163. Please comment on whether the presence of an additional adenine in the S. Typhi sequence (bulging adenine) that is absent in Dublin and Paratyphi C would be expected to affect the thermostability of the secondary structure. Would you expect this to affect thermoregulation in Paratyphi C and Dublin?

8. 809. Please provide information on the source of 30S ribosomal subunit

9. 956-959. A description of how relative expression levels were calculated should be included. Presumably this was the delta delta Ct method?

Reviewer #3: No minor issues detected.

PLOS authors have the option to publish the peer review history of their article (what does this mean?). If published, this will include your full peer review and any attached files.

Reviewer #1: **Yes: **Alexander Westermann

Reviewer #2: No

Reviewer #3: No
---

## [Editor Report · Decision Letter 1]

28 Jan 2021

Dear Denise,

We are pleased to inform you that your manuscript 'A Salmonella Typhi RNA thermosensor regulates virulence factors and innate immune evasion in response to host temperature' has been provisionally accepted for publication in PLOS Pathogens.

Best regards,

Leigh Knodler

Guest Editor

PLOS Pathogens

Renée Tsolis

Section Editor

PLOS Pathogens

Kasturi Haldar

Editor-in-Chief

PLOS Pathogens

orcid.org/0000-0001-5065-158X

Michael Malim

Editor-in-Chief

PLOS Pathogens

orcid.org/0000-0002-7699-2064
---

## [Editor Report · Acceptance letter]

23 Feb 2021

Dear Dr. Monack,

We are delighted to inform you that your manuscript, "A *Salmonella* Typhi RNA thermosensor regulates virulence factors and innate immune evasion in response to host temperature," has been formally accepted for publication in PLOS Pathogens.

Best regards,

Kasturi Haldar

Editor-in-Chief

PLOS Pathogens

orcid.org/0000-0001-5065-158X

Michael Malim

Editor-in-Chief

PLOS Pathogens

orcid.org/0000-0002-7699-2064